# TRAIL-induced variation of cell signaling states provides nonheritable resistance to apoptosis

Reema Baskar[1,2,*], Harris G Fienberg[2,3,*], Zumana Khair[2], Patricia Favaro[2], Sam Kimmey[2,4], Douglas R Green[5], Garry P Nolan[6], Sylvia Plevritis[1], Sean C Bendall[2,3]

**TNFα-related apoptosis-inducing ligand (TRAIL), specifically initiates programmed cell death, but often fails to eradicate all cells, making it an ineffective therapy for cancer. This fractional killing is linked to cellular variation that bulk assays cannot capture. Here, we quantify the diversity in cellular signaling responses to TRAIL, linking it to apoptotic frequency across numerous cell systems with single-cell mass cytometry (CyTOF). Although all cells respond to TRAIL, a variable fraction persists without apoptotic progression. This cell-specific behavior is nonheritable where both the TRAIL-induced signaling responses and frequency of apoptotic resistance remain unaffected by prior exposure. The diversity of signaling states upon exposure is correlated to TRAIL resistance. Concomitantly, constricting the variation in signaling response with kinase inhibitors proportionally decreases TRAIL resistance. Simultaneously, TRAIL-induced de novo translation in resistant cells, when blocked by cycloheximide, abrogated all TRAIL resistance. This work highlights how cell signaling diversity, and subsequent translation response, relates to nonheritable fractional escape from TRAIL-induced apoptosis. This refined view of TRAIL resistance provides new avenues to study death ligands in general.**

## Introduction

Chemotherapeutic drug resistance is one of the critical impediments to successful malignant tumor treatment in humans. Conventional thinking is that a subset of tumor cells variably persists in the face of cytotoxic drugs because of preexisting genetic differences that confer a cell state with a selective survival advantage. However, it has been shown that genetically identical tumor cells display variable cell states that allow differences in response to chemotherapy drug, thereby highlighting a nongenetic basis of resistance that has yet to be extensively explored in human cancers (Brock et al, 2009; Niepel et al, 2009). Variable cell states in tumor cells arise partly because of differential chromatin accessibility, which results in different transcriptomes (Cohen et al, 2008; Shaffer et al, 2017; Litzenburger et al, 2017). Further intercellular differences in translation and degradation drive stochastic differences in the proteome and lead to different cell states despite genetical homogeneity (Brock et al, 2009). In the case of resistance to TNF-related apoptosis-inducing ligand (TRAIL), stochastic variation in levels of proteins involved in apoptosis has been implicated as a nongenetic mechanism of resistance (Spencer et al, 2009; Bertaux et al, 2014).

TRAIL is an endogenous ligand of the TNF family that has been shown to target and induce apoptosis in tumor cells selectively (Wiley et al, 1995; Ashkenazi et al, 1999; Walczak et al, 1999). It binds death receptors (DR4/5) to initiate the formation of death-inducing signaling complexes (DISCs) with the recruitment of adaptor protein FADD (FAS-associated death domain protein) (Kischkel et al, 1995). FADD subsequently activates high levels of pro-caspase 8 and 10 for eventual cell death in type I cells (Boatright et al, 2003; Kantari & Walczak, 2011). In type II cells, additional Bid cleavage and pro-apoptotic Bcl2 family members are required for mitochondrial outer membrane permeabilization and cell death (Özören & El-Deiry, 2002; Rudner et al, 2005). Recombinant TRAIL ligand and monoclonal agonist antibodies to death receptor (DR4/5) were developed as cancer therapeutics but were found to be clinically ineffective, likely because of widespread resistance to TRAIL-induced apoptosis (Herbst et al, 2010; Holland, 2014).

Tumor cells exhibit fractional cell death when exposed to TRAIL, even at saturating levels in vitro, with only a proportion of cells inducing apoptosis (Flusberg et al, 2013; Pavet et al, 2014; Roux et al, 2015). Furthermore, the observed resistance was found to be transient, as tumor cells previously treated with TRAIL demonstrate similar fractional death upon subsequent TRAIL exposure (Spencer et al, 2009; Flusberg et al, 2013). This transient fractional killing is in part explained by the double role of TRAIL in apoptosis canonically as well as in noncanonical, pro-survival, pro-inflammatory, and proliferative signaling (Azijli et al, 2013; Flusberg et al, 2013; Flusberg

[1]Cancer Biology Program, Stanford University School of Medicine, Stanford, CA, USA  [2]Department of Pathology, Stanford University School of Medicine, Stanford, CA, USA  [3]Immunology Program, Stanford University School of Medicine, Stanford, CA, USA  [4]Developmental Biology Program, Stanford University School of Medicine, Stanford, CA, USA  [5]St. Jude Children's Research Hospital, Memphis, TN, USA  [6]Baxter Laboratory, Stanford University School of Medicine, Stanford, CA, USA

Correspondence: bendall@stanford.edu
*Reema Baskar and Harris G Fienberg contributed equally to this work.

& Sorger, 2015; Shlyakhtina et al, 2017). Key transcription factor NF-κB is activated downstream of TRAIL by DISC phosphorylation and subsequent degradation of NF-κB agonist, IκBα (Chaudhary et al, 1997; Jeremias & Debatin, 1998; Ehrhardt et al, 2003; Luo et al, 2005). Other noncanonical signaling pathways such as ERK, Akt, p38, Jnk, and mTOR have been implicated in resistance to TRAIL-induced apoptosis (Azijli et al, 2013; Kim et al, 2000; Lee et al, 2002; Mühlenbeck et al, 1998; Panner et al, 2005; Vaculová et al, 2006; Xu et al, 2010; Zauli et al, 2005). However, much of this work has been carried out in different cancer cell types and bulk populations. The variety of cell states present before and after TRAIL exposure as well as critical signaling modalities that might be conserved across cancer types has yet to be interrogated.

In addition to survival pathway activation, fractional killing by TRAIL has also been explained by variation in pro and anti-apoptotic protein abundance in tumor cells because of genetic aberrations or nongenetic mechanisms (Zhang & Fang, 2005). Studies characterized nongenetic mechanisms of variation in levels of anti-apoptotic protein cellular FLICE (c-FLIP), which binds to FADD and prevents formation of DISC, in spatial organization of DR4/5 and in rate and duration of initiator caspase activation by DISC (French et al, 2015; Marconi et al, 2013; Roux et al, 2015; Twomey et al, 2015). Recent work indicates that multiple mechanisms might be at play simultaneously to result in intercellular variation and, therefore, cell states of resistance (Ramirez et al, 2016; Salgia & Kulkarni, 2018). This underlines the need for a single-cell approach to capture multi-dimensional cell states. New single-cell proteomic technologies, such as mass cytometry, allow us to identify apoptotic and survival cell states in a multiplexed and high-throughput fashion upon TRAIL exposure and analyze the intercellular variation that might be driving resistance (Bendall et al, 2011; Fienberg & Nolan, 2014).

To that end, we interrogated TRAIL responsiveness across 10 human cell lines, representing a multitude of cell lineages, under several experimental conditions; followed by simultaneous, single-cell interrogation of canonical and noncanonical apoptotic and survival signaling pathways downstream of TRAIL with mass cytometry. This enabled us to probe multiple layers of the cellular response to TRAIL, including tracking of crucial kinase signal transduction, apoptotic induction, cell cycle status, and de novo protein synthesis, as they relate to apoptotic escape. We show that each tumor cell line has a different level of fractional killing and surprisingly have distinct yet conserved signaling responses to TRAIL. We highlight the role of cell-to-cell variation in the survival signaling pathways in TRAIL resistance for the first time, paralleling variation found in the canonical apoptotic induction pathways (Spencer et al, 2009). We also show for the first time that the abundance of TRAIL receptor 1 (death receptor 4 [DR4]) on the cell surface before TRAIL exposure could dictate noncanonical response and, therefore, resistance to TRAIL across cell lines.

Cellular heterogeneity of tumors and its involvement in developing resistance to chemotherapies is known, but we are yet to quantify it robustly and decode the underlying conserved mechanisms that are driving this cell-to-cell variation (Fallahi-Sichani et al, 2013; Sun & Yu, 2015). To quantify this, we used Shannon diversity index as a multidimensional signaling heterogeneity metric to analyze the effect of TRAIL on signaling states in cancer cells. We then altered the signaling diversity by reducing the

permissible signaling states of resistance using inhibitors to critical signaling pathways during TRAIL treatment. Importantly, we found that the altered signaling diversity inversely correlates significantly with level of resistance to different combination therapies. Furthermore, through simultaneous application of a novel single-cell translation assay, we found TRAIL-induced de novo protein synthesis in these same resistant cells. A specific, short-term blockade of this with a sub-cytotoxic dose of cycloheximide could effectively abrogate all TRAIL resistance, resulting in apoptosis of all cells treated. Collectively, this work highlights the integral role of variation in a noncanonical signaling pathway that seems to lead to translation of protein factors that contribute to a nongenetic mechanism of resistance to TRAIL-induced apoptosis. This lays the groundwork for building comprehensive models of TRAIL resistance which incorporates single-cell readouts of apoptosis, survival signaling, and cell physiology activity.

# Results

### Cells variably persist in TRAIL across cancer lines

TRAIL-induced apoptosis progresses asynchronously over hours, engaging the canonical external apoptotic pathway as well as the noncanonical survival signaling pathways (Flusberg & Sorger, 2015; Shlyakhtina et al, 2017). It is also known to interact with the cell cycle processes of tumor cells (Ehrhardt et al, 2012, 2013). To gain a systems-level perspective on TRAIL resistance, we treated 10 cancer cell lines representing varied cancer types with TRAIL and analyzed their single-cell responses across all three processes using a panel of 30 markers covering apoptotic response, signaling response, cell cycle status, and translation rate with mass cytometry (Fig 1A and Table S1) (Bendall et al, 2011).

We profiled the apoptotic status of tumor cells with antibodies to cleaved caspase 3 (cCasp3) and cleaved poly (ADP-ribose) polymerase (cPARP). In HeLa C9 and NCIH460 cell lines, the percentage of viable, non-apoptotic cells decreased with increasing exposure to TRAIL (Figs 1B and S1A and B). Most viable cells moved out of the non-apoptotic gate (cCasp3$^{low}$ and cPARP$^{low}$), whereas a proportion of cells remained and exhibited the fractional killing phenomenon (Figs 1B and S1A and B, black arrow). Interestingly, even at the level of apoptotic reporters, HeLa C9 and NCIH460 cells have different patterns of progression with HeLa C9 cells having a more continuous increase in cCasp3 and cPARP (Figs 1B and S1B). Even with this trend, the proportion of non-apoptotic cells levels out after reaching a threshold around 5–7 h, despite continued TRAIL exposure till 24 h (Fig 1C, black line). Moreover, this phenomenon of fractional killing has even been observed with increasing concentration of TRAIL (Flusberg et al, 2013). Fractional killing occurs despite altering the treatment duration or dosage of TRAIL which hints at an underlying conserved mechanism of resistance at play.

To better understand what might be underlying this phenomenon, we investigated the level of fractional killing by TRAIL across cancer types with 10 different cancer cell lines (Fig 1D). We calculated a survivor quotient (SQ, ratio of percentage of non-apoptotic cells after TRAIL treatment to the percentage before treatment) to

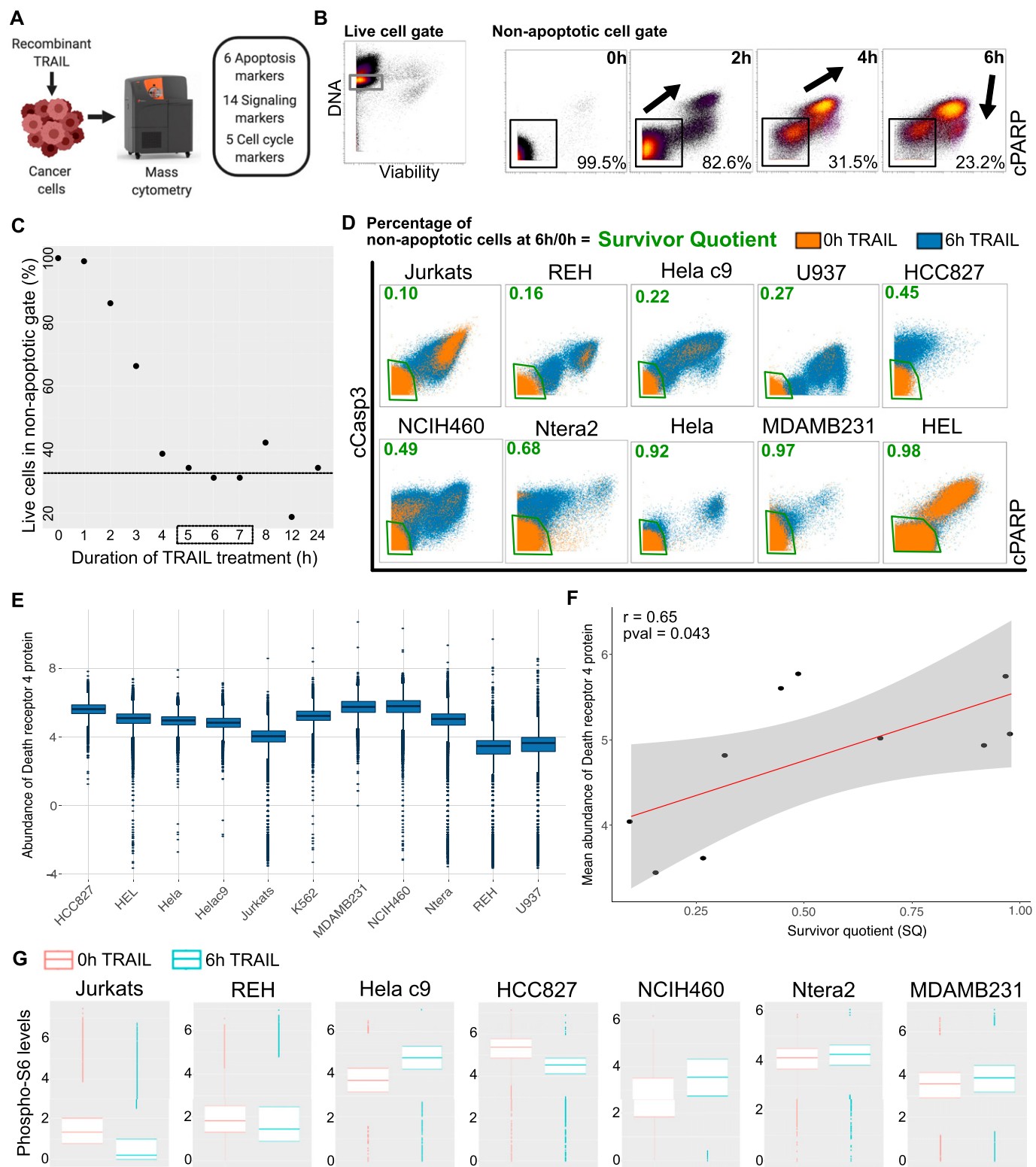

**Figure 1. Cell type–specific persistence in TRAIL-induced apoptosis.**
**(A)** Overview of experimental pipeline. 10 different cancer cell lines representing different cancer types were treated with 50 ng/ml recombinant TRAIL and harvested for analysis on mass cytometry after intracellular staining with metal isotope-conjugated antibodies to apoptotic, survival signaling, and cell cycle proteins. **(B)** Live HeLa C9 cells were gated based on cisplatin for live/dead staining (Fienberg et al, 2012). Canonical apoptosis markers, cleaved caspase 3 (cCasp3), and cPARP were captured on live HeLa C9 cells with increasing exposure to TRAIL. The percentage of live cells in the black non-apoptotic gate is indicated. The black arrows visualize direction of movement of population with time. **(C)** The percentage of non-apoptotic HeLa C9 cells (cCasp3$^{low}$ and cPARP$^{low}$) is tracked from 0–24 h of treatment with TRAIL. The dotted

quantify the level of resistance to TRAIL-induced apoptosis after 6 h of TRAIL treatment. The different cancer types have highly varying levels of apoptotic resistance, as signified by the varying SQ (Fig 1D, *green value*). Previous work suggested that endogenous preexisting levels of apoptotic regulators, such as DR4 and c-FLIP, might explain the differential levels of resistance (French et al, 2015; Li et al, 2011; Zang et al, 2014; Zong et al, 2009).

To explore this possibility, we captured the abundance of DR4 protein on the cell surface membrane of 11 endogenous cell lines using flow cytometry after validating the antibody on mRNA level-based positive and negative cell line controls (Figs 1E, and S1E and F). The level of DR4 differed between cell lines and did not show high variation within each cell line (Figs 1E and S1F). We correlated the level of DR4 protein to our calculated SQ across cell lines and interestingly found a significant positive correlation indicating that higher levels of DR4 corresponded to greater resistance to TRAIL-induced apoptosis (Fig 1F). Similarly, we show a positive correlation between SQ and mRNA levels of TNFRSF10A, given mRNA and protein abundance levels of DR4 are highly correlated (Fig S1C and D). This strongly highlights the need to interrogate the noncanonical response to TRAIL binding DR4 as it results in a non-apoptotic cellular response over apoptosis.

All the 10 cell lines we tested have a proportion of viable, non-apoptotic cells remaining after 6 h of TRAIL treatment (Fig 1D). However, like the pattern of the apoptotic factors cCasp3 and cPARP, these resistant cells did not respond similarly to TRAIL with respect to regulatory cell signaling (Fig 1E). For example, phosphorylation of ribosomal protein S6 is a known mediator of TRAIL signaling and works in the mTOR pathway to primarily regulate translation (Jeon et al, 2008; Panner et al, 2005; Sridharan and Basu, 2011). It has been implicated in other drug resistance phenomenon as well and highlights the role of translation in acquisition of resistance (Sun et al, 2014; Gao et al, 2018). Levels of phosphorylated S6 (pS6) change with TRAIL exposure differentially between cell lines (Fig 1G). This cell type–specific fractional killing combined with DR4 correlating to resistance and differential pS6 response to TRAIL underlines the need to more broadly assay signaling responses to TRAIL and the relationship to apoptotic escape.

### Repeat TRAIL signaling responses are consistent and resistance is not heritable

Genetic aberrations that confer resistance is selected for by drug and persists heritably through cell divisions (Klein, 2009; Hu & Zhang, 2016; Salgia & Kulkarni, 2018). The involvement of nongenetic mechanisms is revealed by the inability to sustain the resistant phenotype. Previous work shows that the acquired resistant cell state cannot persist beyond immediate re-treatment (under 2 d) with cells resetting and exhibiting similar fractional killing and transcriptome as cells with no prior exposure to TRAIL (Flusberg et al, 2013). We built upon these findings by re-treating two known

resistant cell lines of diverse cancer types of cervical and lung cancer, HeLa C9, and NCIH460, with TRAIL 10 d after an initial treatment, followed by mass cytometry analysis (Fig 2A, *top*). Cells from the two cell lines exhibit similar levels of apoptotic induction regardless of prior exposure to TRAIL (Fig 2A, *bottom*). Furthermore, the resistance potential of a cancer type is conserved across biological replicates as shown by the conserved frequency of resistant colonies in clonal plating assays (Figs 2B and S2A).

To investigate how regulatory signaling induced by TRAIL changes with repeat exposure and between cell lines, we built a mass cytometry panel consisting of known apoptotic proteins and noncanonical signaling proteins implicated downstream of TRAIL activation to quantitatively capture behavior in single resistant cells (Table S1 and Fig 2C). We probed the signaling response of cells to TRAIL before and after the 10 d "drug holiday" (Fig 2A, *top*) and show that they have highly correlated signaling profiles (Figs 2D and S2C) after repeat exposure. This lack of a differential signaling memory in TRAIL-resistant cells aligns with the similar apoptotic and survival cellular responses in these conditions. Importantly, all non-apoptotic cells, regardless of prior TRAIL exposure, signal in response to the ligand considering all signaling markers by t-stochastic neighbor embedding (Fig S2D). However, we note that this reversible, transient resistant signaling state is not entirely stochastic and stays consistent across biological replicates (Fig S2B). The conservation of a resistant signaling response to TRAIL, regardless of prior exposure, hints at an underlying priming toward an induced cell survival state previously described in other drug resistance phenomena (Brown et al, 2014; Smith et al, 2016). Understanding this state could be key to mitigating apoptotic resilience.

### TRAIL induces cell cycle changes which do not persist or correlate significantly with resistance

Outside of regulatory cell signaling, another mechanism of drug resistance can be changes in cell cycle, such as arresting in G1 phase or moving to G0 senescence (Quast et al, 2015; Beaumont et al, 2016). To investigate the effect of cell cycle on TRAIL resistance, we gated non-apoptotic cells into the different cell cycle phases and mapped state changes across a TRAIL treatment time course. In HeLa C9 cells, we observed changes to cell cycle frequencies with an increase in G0 cells with a proportional decrease in the S-phase. This was followed by a return to pretreatment like cell cycle frequencies by 24 h (Figs 3A and S3A) (Behbehani et al, 2012). Interestingly, previous work has found TRAIL-mediated proliferation in resistant cells and higher sensitivity to TRAIL in cell cycle-arrested cells in different cancer types (Ehrhardt et al, 2013). Here, the cells adjust to TRAIL exposure and begin to recover their original cell cycle distributions after 7 h or more of continued TRAIL exposure (Fig 3A). This time dependency in TRAIL response is seen for the first time in our work here and highlights the importance of

---

line indicates a plateau in the fractional killing proportion in the cell line. **(D)** The live cells at 0 and 6 h of TRAIL treatment is overlaid in dot plots across 10 cell lines with the percentage of non-apoptotic cells at each time point quantified to calculate the SQ ratio metric. **(E)** Boxplots show arcsinh-transformed levels of DR4 across cell lines. **(F)** Dot plot shows mean abundance of DR4 and SQ across cell lines with Pearson correlation between them. **(G)** Boxplots of phosphorylated S6 levels in equally subsampled, untreated and 6 h TRAIL-treated non-apoptotic cells across cell lines.

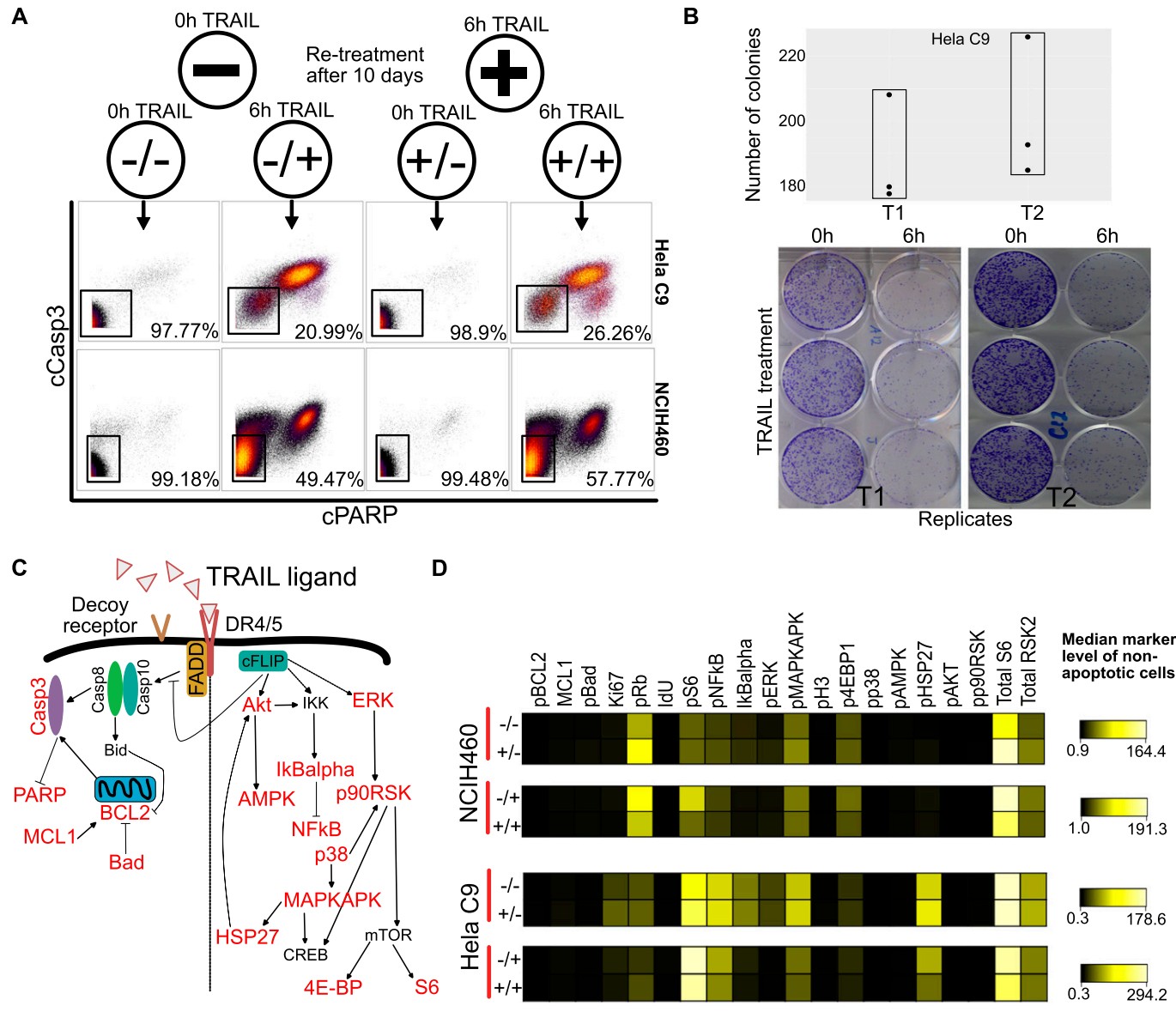

**Figure 2.  Apoptotic resistance is not heritable and TRAIL signaling is reset with each exposure.**
**(A)** Overview of TRAIL re-treatment experiment and dot plots of different re-treatment conditions. The percentage of viable cells in the black non-apoptotic gate is as shown. **(B)** Clonogenic analysis of resistance by HeLa C9 cells to TRAIL across three technical replicates and two biological replicates (T1 and T2). **(C)** Schematic of known apoptotic and survival signaling pathways downstream of TRAIL receptor. Proteins captured for mass cytometry analysis are in red. **(D)** Heat map of median abundance of markers between treated and re-treated samples. Red line indicates significant correlation between values.

appropriate exposure lengths to TRAIL for maximal effect before stable resistance sets in.

At the same time, a different cancer cell line, HCC827, exhibited different cell cycle dynamics upon TRAIL exposure as cells moved out of the G1 state (Fig S3B). To more broadly assess this, we looked at TRAIL-induced cell cycle changes across all 10 cell lines (Fig 3B). We found that TRAIL-induced cell cycle changes are cell line dependent with different cell lines increasing in specific cell cycle phases such as G0 phase for NCIH460, HCC827, and Jurkats; G1 phase for MDAMB231 and Ntera; and M phase for REH and HeLa C9 (Fig 3B). TRAIL-induced changes in each cell cycle phase as well as the combined changes across all phases did not correlate with the SQ

of cell lines (data not shown). Consequently, although TRAIL seems to have selective effects on cell cycle phase, it appears to be cell type specific without conserved trends and not related to overall apoptotic resistance of cells.

## Resistant cells persist in a TRAIL-induced signaling state

In the paradigm of preexisting genetic aberrations, resistant cells are selected for by being refractory (i.e., nonresponsive) to a drug. In acquired resistance, generally, a subset of cells responds and moves to the resistant state, whereas the rest move to an apoptotic state before dying. For the first time, we show evidence for an

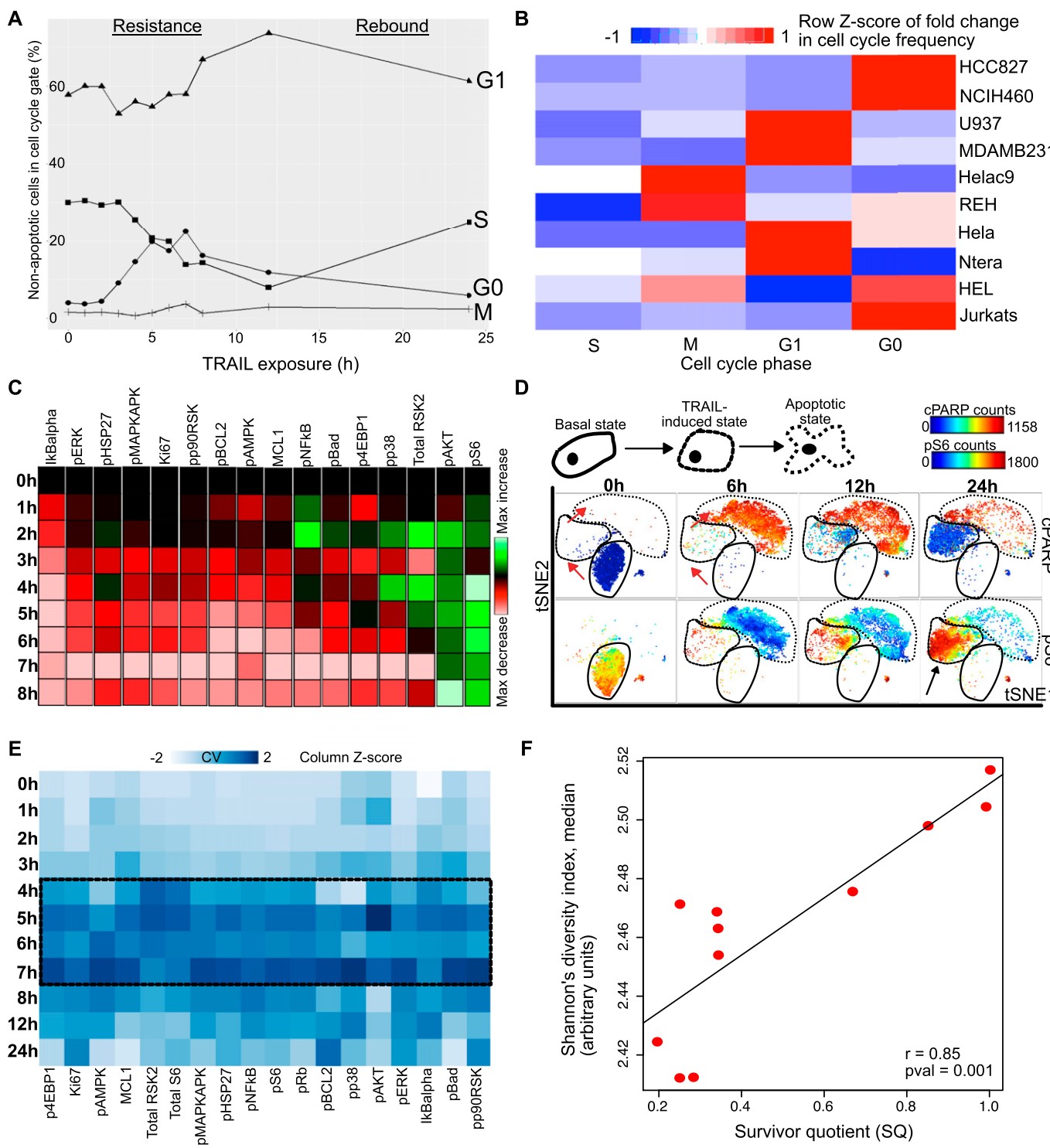

**Figure 3. All cells respond but resistant cells persist in a TRAIL-induced signaling state.**
**(A)** Line graph of four cell cycle phases tracking percentage of viable HeLa C9 cells in the four cell cycle phases from 0 to 24 h of TRAIL exposure. **(B)** Heat map of fold change in percentage of non-apoptotic cells in cell cycle gate between untreated and 6 h TRAIL treatment scaled by column. **(C)** Heat map is individually scaled column-wise and colored by the arcsinh ratio relative to 0-h treatment to illustrate the increase or decrease of marker relative to basal state in non-apoptotic HeLa C9 cells. **(D)** viSNE representation of 20 K equally subsampled signaling states of cells moving from basal, through TRAIL-induced state to apoptotic state across time points (red arrows). Maps are colored by cPARP and phosphorylated S6 levels. Black arrow indicates cells that persist in the TRAIL-induced state. **(E)** Heat map colored by coefficient of variation of individual markers in equally subsampled, non-apoptotic HeLa C9 cells from 0 to 24 h of TRAIL exposure (individually scaled column-wise). **(F)** Dot plot shows mean signaling diversity and SQ across time course of TRAIL treatment and Pearson correlation between them.

alternative trajectory of acquired resistance where all cells respond and move to a survival signaling state, but only a subset persists in it over time and remains resistant with the remaining cells progressing to an apoptotic state (Fig S3E).

We captured the single-cell responses to increasing exposure of TRAIL with our mass cytometry panel and charted the signaling differences from untreated cells (Table S1 and Fig 3C). Many of the survival signaling proteins have increased activity with higher levels of phosphorylation after 2 h of TRAIL exposure (Fig 3B). The higher abundance of RSK2 kinase and phosphorylated p38 kinase is not sustained with increasing exposure to TRAIL (Fig 3C). The RSK2 kinase is a known modulator of S6 kinase activity and has been implicated in TRAIL-induced apoptosis through its phosphorylation and subsequent degradation of pro-apoptotic protein caspase 8 (Panner et al, 2005; Peng et al, 2011). Although RSK2 might be crucial in mounting resistance to initial exposure to TRAIL, as the cells tolerize to TRAIL and persist with time, it is likely not required to maintain the resistant signaling state (Fig S3E). Similarly, activated p38 kinase is a crucial survival signaling protein that is known to increase within 2 h of TRAIL exposure, but we show that this does not persist with more prolonged exposure (Fig 3C) (Piras et al, 2011). Only two survival signaling proteins, AKT and S6 kinases, are up-regulated and show higher than basal levels of activation after 5 h of TRAIL exposure (Fig 3B). This supports the known role of AKT and its regulation of mTOR and, therefore, S6 kinase in the maintenance of the TRAIL resistance state (Xu et al, 2010). Other signaling proteins, such as NFkB, have dual roles in TRAIL resistance. It has been shown that NFkB not only promotes death receptor expression but also increases expression of apoptosis inhibitor Bcl-xL (Ravi et al, 2001; Luo et al, 2005). The specific levels and activity of the signaling and apoptotic markers we captured are essential in modulating anti- or pro-apoptotic responses to TRAIL.

We mapped the high-dimensional signaling response to TRAIL into lower dimensions using t-stochastic neighbor embedding to visualize the different cellular states and the frequency of cells in each state with increasing exposure to TRAIL (Figs 3D and S3D) (Amir et al, 2013). We identified three distinct states from our signaling markers: a basal state, a TRAIL-induced state, and an apoptotic state (Figs 3D and S3D). The time course here demonstrates that virtually all cells respond to TRAIL by moving from the basal state to a TRAIL-induced signaling state (Figs 3D and S3D). However, this state transition is not immediate; the cells only move to the TRAIL-induced signaling state around 4 h, and are likely tolerized to TRAIL after 7 h, which is reflected in cell cycle changes and marker variance (Fig 3A and E). It is from this intermediate state that a proportion of cells persist and do not progress to apoptosis, even with continued TRAIL exposure (Fig 3D, *black arrow*). This latter transition is likely dependent on a caspase cleavage cascade in late apoptosis activation that we further confirmed by applying a caspase inhibitor which prevented most cells from undergoing this terminal process (Fig S3C). Our ability to inhibit death by TRAIL indicated a potentially ordered process of cellular response to TRAIL; therefore, we sought to model the cellular response to TRAIL as a trajectory.

Our results suggested a linear progression from a basal to TRAIL-induced signaling state where cells either persist or proceed to apoptosis (Figs 3D and S3D). We captured the signaling progression from basal to TRAIL-induced resistant state by building a linear,

pseudo-time trajectory of non-apoptotic cells across time points with the Wanderlust algorithm (Fig S3E and F) (Bendall et al, 2014a). The constructed trajectory revealed the rapid transitions that cells underwent in specific signaling proteins and showed the specific states that cells remained in without progressing to apoptosis (Fig S3D and E). Given the asynchrony of apoptotic progression, this trend was not surprisingly hidden within time-based fold change analysis on bulk populations (Fig 3C) (Spencer et al, 2009). This not only highlights the need for high-throughput single-cell techniques to model such processes, but also further emphasizes the previously overlooked role of noncanonical signaling in resistance to TRAIL-induced apoptosis.

We see a similar pattern of time dependency seen in the cell cycle and signaling response in the variance of markers across the TRAIL time course (Figs 3A, S3D, and 3E). Variation is known to be implicated in nongenetic mechanisms of resistance to TRAIL (Spencer et al, 2009; Frank & Rosner, 2012). Signaling protein abundances as well as their variation changed with exposure to TRAIL (Fig 3C and E). The signaling markers are relatively homogenous at short exposures. However, with increasing exposure, the variation in signaling markers also increases and then decreases towards baseline at 24 h. The highest variation across all markers is seen around 4–7 h of TRAIL exposure and coincides with stabilization of the frequency of TRAIL-resistant cells and full shift to the TRAIL-induced signaling state (Figs 1C and 3E, and S3D). We calculated the variation of individual signaling proteins using coefficient of variation and we calculated the overall signaling state diversity with Shannon diversity index on unit-scaled arcsinh-transformed signaling marker counts. We saw a highly significant positive correlation between resistance (SQ) and diversity of signaling states in non-apoptotic cells (Fig 3F). Variation is likely playing a role in the noncanonical survival signaling response and, therefore, resistance to TRAIL-induced apoptosis.

## Signaling diversity in response to TRAIL correlates with resistance to apoptosis

To identify conserved mechanisms of resistance in the signaling response to TRAIL, we expanded our analysis of single-cell TRAIL responses at 6 h to 10 cell lines and compared the fold change in signaling markers from untreated condition (Fig 4B). Much like the cell cycle behavior in response to TRAIL, the signaling responses were fairly cell line specific and with some conservation in IkBalpha and pS6 (Fig 4B). As expected, based on canonical TRAIL signaling (Fig 2C), most cell lines generally increased NFkB through decreasing its inhibitor IkBalpha as well as up-regulating phosphorylated S6, presumably through upstream mTOR or AKT activation (Fig 4B).

To more broadly identify signaling modalities in TRAIL-resistant cells, we clustered non-apoptotic cells after 6 h of TRAIL treatment and found eight different signaling states (Fig S4A, *left*). These clusters were similar in levels of p4EBP1, pS6, pMAPKAPK, and total S6, but each cluster was also generally dominated by a single cell line (Fig S4A). The resistant state with the most substantial number of cells (cluster 1) is made up of a few cell lines such as HeLa C9, HCC827, and HeLa and has relatively high levels in more than half of the signaling markers (Fig S4A, *right*). However, no signaling state was present across all TRAIL-sensitive cell lines that represented a conserved state of resistance from which one could deduce a mechanism pertaining to one or more signaling proteins.

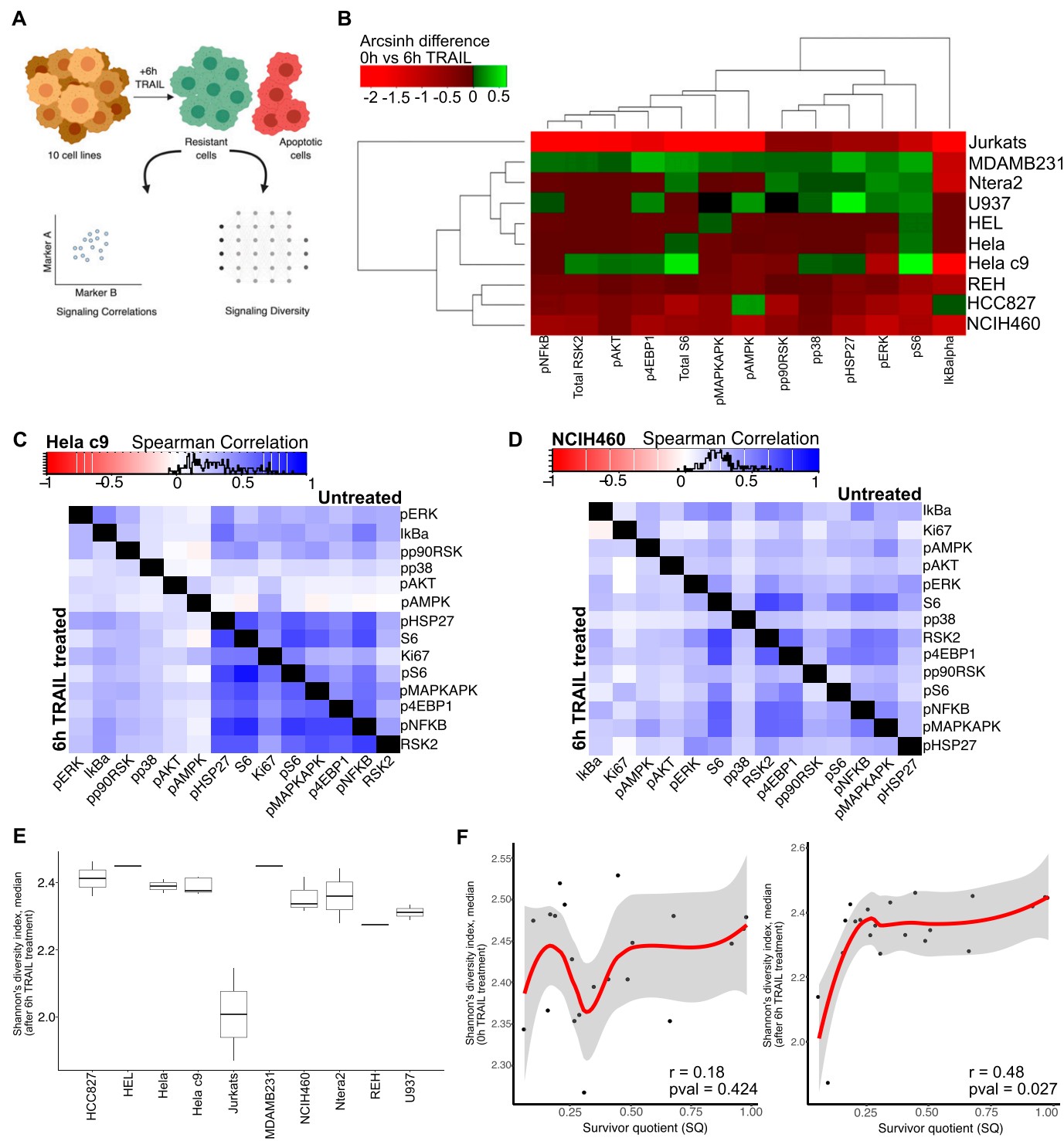

**Figure 4. TRAIL-induced change in cell signaling entropy correlates with apoptotic resistance across cell types.**
**(A)** Schematic of single-cell analysis carried out across cell lines **(B)** Heat map of arcsinh fold changes of median abundances of signaling marker between 0 and 6 h of TRAIL-treated non-apoptotic cells. Heat map rows and columns ordered by hierarchal clustering. **(C)** Differential Pearson correlation between signaling marker abundance in untreated and 6-h TRAIL–treated non-apoptotic HeLa C9 cells. **(D)** Differential Pearson correlation between signaling marker abundance in untreated and 6-h TRAIL–treated non-apoptotic NCIH460 cells. **(E)** Boxplots showing median signaling diversity calculated over 14 markers (Table S1) across cell lines (with 1–4 biological replicates). **(F)** Dot plot showing median signaling diversity and SQ across cell lines in untreated cells (left) and in 6-h TRAIL–treated cells (right) with Pearson correlation coefficient and *P* value displayed.

With this in mind, we sought to quantify signaling diversity and understand the relationships between the signaling markers in our assayed cell lines to find what might correlate with the SQ of cell lines from our mass cytometry readout of multiplexed, single-cell signaling response to TRAIL (Fig 4A). Cell lines did not drastically differ in their signaling marker correlations before and after TRAIL treatments in individual lines; however, across cell lines, they do show differences in marker correlations (Figs 4C and D, S4B–E, and S5A–D). Phosphorylated ERK (extracellular signal-regulated kinase), IkBalpha, and P90RSK show stronger positive correlations with many of the other signaling markers after 6 h of TRAIL treatment (Fig 4A). Signaling proteins in NCIH460 show different correlation patterns as can be seen from the different hierarchal arrangement of markers (Fig 4D). There is no conserved pattern of signaling protein relationship across cell lines to identify a universal signaling modality in TRAIL resistance.

Therefore, we hypothesized that the overall cell signaling heterogeneity in the noncanonical response to TRAIL might be important to resistance (Spencer et al, 2009; Roux et al, 2015). We quantified the signaling diversity in non-apoptotic cells after 6 h of TRAIL resistance across our 10 cell lines (Fig 4E). Different cell lines showed different resulting diversity upon TRAIL exposure, with Jurkats showing the lowest diversity (Fig 4E). Specifically, signaling diversity after TRAIL treatment and not before correlated significantly with resistance to TRAIL (SQ), showing that TRAIL might be inducing signaling diversity (Fig 4F). There is a likely a nonlinear relationship hinting at a resistance threshold that once reached, cells have similar signaling diversities regardless of cell type (Fig 4F). Therefore, it is specifically the induction of variation by TRAIL that underlies the acquisition of resistance where the higher the diversity of signaling states induced by TRAIL, the higher the ability of the population to persist in the resistant state without progressing to apoptosis.

## Constricting signaling space of TRAIL responses decreases apoptotic resistance

We were able to relate TRAIL-induced signaling entropy to apoptotic resistance and, therefore, hypothesized that constricting the diversity in survival pathway responses to TRAIL would lead to more cells progressing to apoptosis. To restrict permissible signaling states, we combined TRAIL with various kinase inhibitors to critical signaling proteins downstream of TRAIL activation (Fig 5A). Our combination of inhibitors changed the fractional killing frequency of cells and resulted in similar or lower SQs (Fig 5B and C, *in green*). Interestingly, the signaling response of two cell lines to the combination treatments of TRAIL and inhibitors differed dramatically with HeLa C9 showing a much stronger up-regulation of signaling than NCIH460 (Figs 5D and S6A). Inhibitors to p38, PI3K, and mammalian target of rapamycin (mTOR), showed compensatory signaling changes in AKT, AMPK, and HSP27 and expected reduction in pAKT with PI3K inhibition and reduction in pS6 with mTOR inhibition (Figs 5D and S6A). These complex changes in signaling modalities in response to combination TRAIL and inhibitor treatments create challenges in predicting resistant states and level of fractional killing. As such, population-level characteristics of single-cell abundances such as individual marker variation and high-dimensional signaling diversity offer a more robust assessment of cellular responses and prediction of apoptotic response to inhibitor combinations.

Here, we observe that the combination treatments on two cell lines, HeLa C9 and NCIH460, change the marker variation by restricting permissible signaling state cells that can occupy when only exposed to TRAIL (Figs 5B, E, and C, and S6B for each cell line respectively). Cells are forced to occupy different signaling states, and markers show higher variation especially with P38 and JNK inhibitors (Fig 5B and E). However, when we look at the overall signaling diversity across markers and in two cell lines, we see a significant positive correlation between diversity induced by combination therapy and the resulting difference in resistance (Fig 5F). This is in line with our previous observation that the induction of signaling diversity by TRAIL is linked to greater resistance to apoptosis (Figs 4F and 5F). Overall, these data demonstrate that signaling diversity could be used as a metric to quantify permissible cell state space of resistance and, therefore, predict the level of fractional killing and efficacy of combination inhibition (Cheng et al, 2016).

We further dug into the relationship between signaling proteins by looking at the differential correlation between cells treated with only TRAIL or with combination therapy (Figs 5G and S6C–E). Interestingly, phosphorylated IkBalpha and AMP-activated protein kinase (AMPK) showed the most different correlations with mTOR inhibition (Fig 5G). There was an increase in mutual information between phosphorylated IkBalpha and nuclear factor kappa-light-chain-enhancer of activated B cells (NFkB), indicating that cells are signaling more strongly through the NFkB pathway and less strongly between phosphorylated AMPK and NFkB in response to mTOR inhibition (Fig 5H). We are able to broadly capture signaling protein relationships and signaling pathway flux with our high-dimensional signal cell data to better understand combination treatment with TRAIL.

## TRAIL signaling induced de novo translation is correlated with increased signaling and apoptotic escape

Because ribosomal protein S6 phosphorylation seen here (Fig 1G) is an indicator of translational activity, we hypothesized that differences in translation rate might be linked to the diversity of signaling states and contribute to TRAIL resistance. Previous work has shown the efficacy of treating cancer cells with a combination of TRAIL and the protein synthesis inhibitor, cycloheximide, to maximize the apoptotic response (Mori et al, 2005; Spencer et al, 2009). Here, we treated the cells with cycloheximide at a subcytotoxic dose with or without TRAIL. By itself, cycloheximide did not induce apoptosis, but when combined with TRAIL, it resulted in almost no surviving cells (Figs 6A and S7A) (Mori et al, 2005; Spencer et al, 2009).

To understand the relationship between protein synthesis and resistance to TRAIL-induced apoptosis across cell lines and combinatorial inhibition, we simultaneously captured the relative rate of de novo translation using a novel single-cell reporter in our panel of signaling and apoptotic markers (see the Materials and Methods section) (Kimmey et al, 2019). Interestingly, this revealed that the TRAIL-responsive, yet resistant, cells exhibited high levels of de novo translation, which could be abrogated by additional cycloheximide (Figs 6B and S7B, *black arrows*). Similarly, the cells that remained non-apoptotic after kinase inhibition also exhibited high rates of translation (Fig S7C and D, *black circle*) even in cases where combination treatment with kinase inhibitor has a protective effect from TRAIL (i.e., mTOR inhibition). Untreated cells do not have as high puromycin levels as compared with TRAIL or combination therapy treated cells

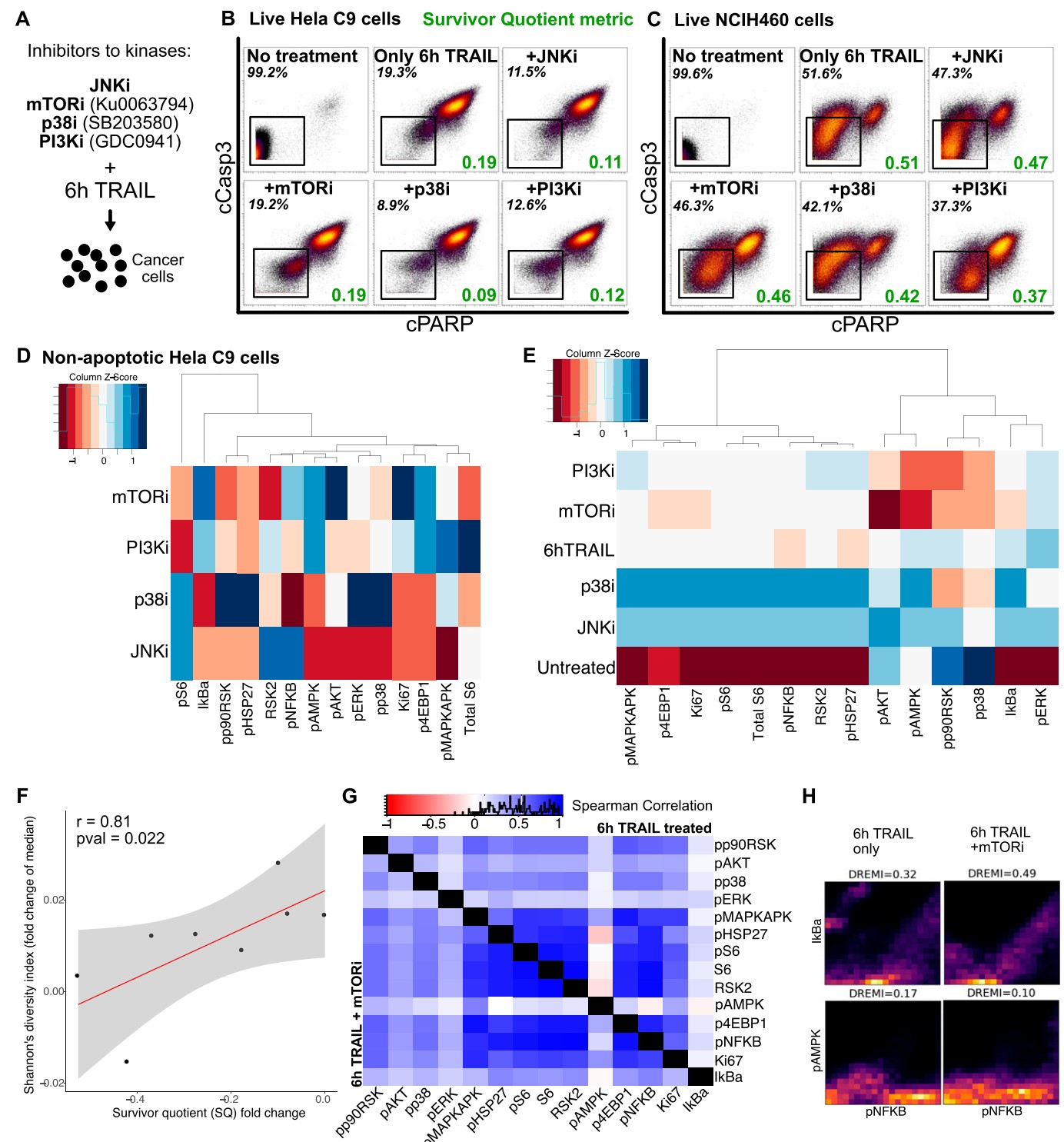

**Figure 5.  Combination therapies decrease resistance to TRAIL by restricting signaling diversity.**
**(A)** Overview of experiment to test combination therapy of TRAIL with kinase inhibitors targeted at key survival signaling proteins. **(B)** Dot plots showing apoptotic induction in viable HeLa C9 cells with cCasp3 and cPARP with combination therapies in HeLa C9 cells and SQ shown in green. **(C)** Dot plots showing apoptotic induction in viable NCIH460 cells with cCasp3 and cPARP with combination therapies in HeLa C9 cells and SQ shown in green. **(D)** Heat map of arcsinh fold change of median signaling marker abundance in non-apoptotic HeLa C9 cells between combination treatments and TRAIL treatment only scaled by markers. **(E)** Heat map signaling marker variation (calculated by coefficient of variation) in non-apoptotic HeLa C9 cells scaled column-wise by markers. **(F)** Dot plot showing fold change in median signaling diversity and fold change in SQ between only TRAIL and combination therapy treatments with Pearson correlation calculated between them. **(G)** Differential correlation of signaling markers between 6-h TRAIL treatment and mTOR inhibitor combination therapy in non-apoptotic HeLa C9 cells. **(H)** Top differentially correlated marker pairs from Fig 5G visualized on conditional-Density Resampled Estimate of Mutual Information (DREMI) plots with their scaled mutual information shown.

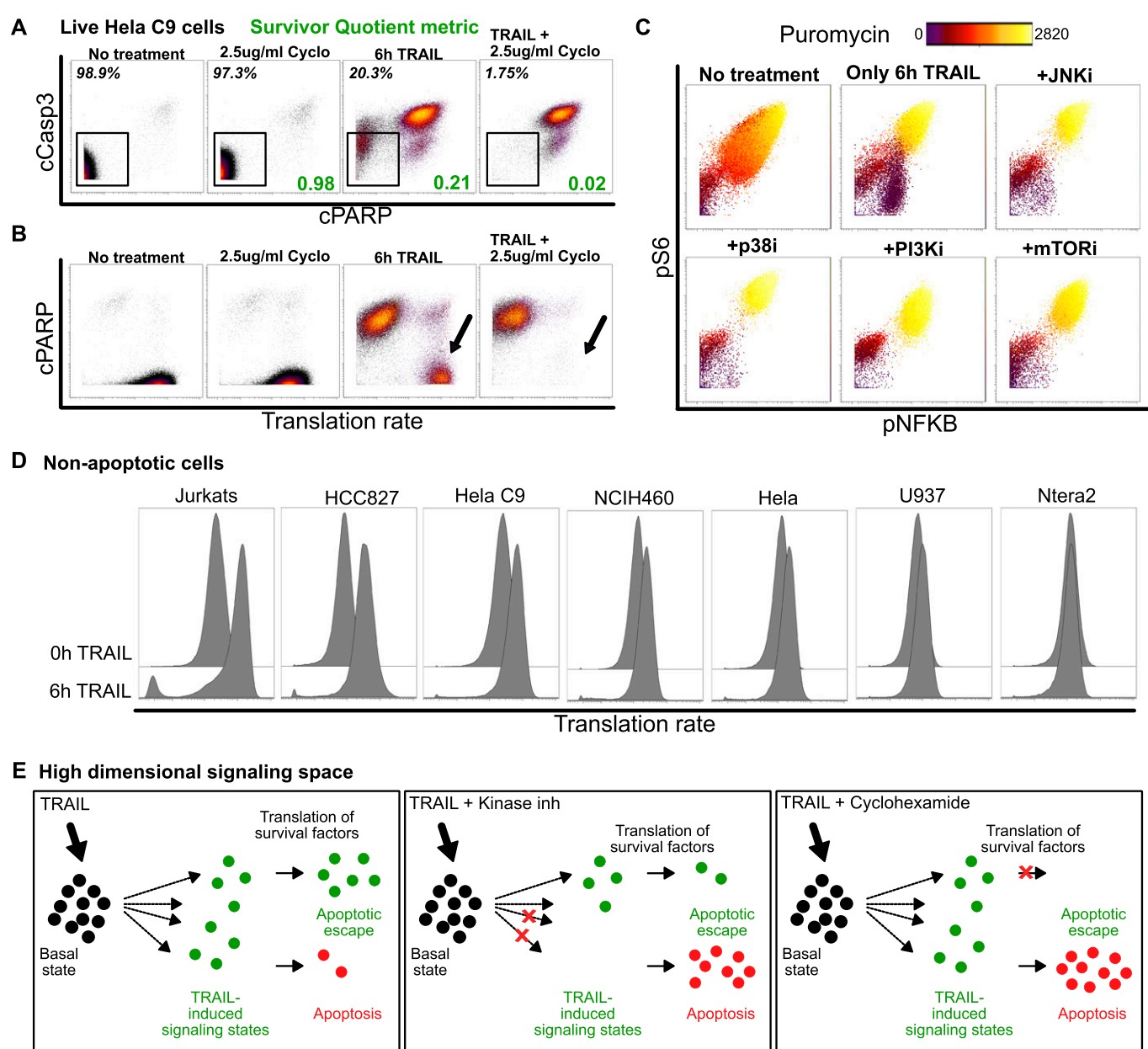

**Figure 6. TRAIL-induced increase in translation is linked to apoptotic resistance.**
**(A)** Dot plots showing apoptotic induction (percentage of live, viable cells, in black) in HeLa C9 cells with combination therapy of cycloheximide (protein synthesis inhibitor) and TRAIL (SQ shown in green). **(B)** Dot plots showing puromycin levels that track single-cell translation rate in viable HeLa C9 cells with combination therapy of cycloheximide and TRAIL. **(C)** Dot plots showing phosphorylated S6 and NFkB in non-apoptotic HeLa C9 cells across combination inhibitor treatments with puromycin levels in color overlay. **(D)** Histograms of puromycin levels in non-apoptotic cells across cell lines. **(E)** Summary schematic of paper showing role of signaling diversity and translation in resistance to TRAIL-induced apoptosis.

suggesting TRAIL treatment was inducing an increase in translation through the noncanonical signaling response (Fig 6B and C). We see that the resistant cells have distinct low and high translating populations that correspondingly have low and high signaling markers across combination therapies suggesting a likely feedback loop between translation and these key signaling proteins (Figs 6C and S7E).

Furthermore, particularly in cell lines more sensitive to TRAIL (i.e., Jurkat and HeLa C9), translation rates are much more increased in the non-apoptotic cells after TRAIL treatment (Fig 6D). These data illustrate how translation could be producing protein survival factors that further increase translation to produce a positive feedback loop that keeps cells in the noncanonical response and evading TRAIL-induced apoptosis. Overall, we examine TRAIL response in 10 different cell lines and show how TRAIL-induced signaling diversity and translation correspond to resistance to TRAIL-induced apoptosis (Fig 6E).

# Discussion

Here, we highlight the role of variation and protein synthesis in the noncanonical signaling response to TRAIL and how it relates to escape from TRAIL-induced apoptosis (Fig 6E). We demonstrate that despite uniform activation across TRAIL-responsive cell lines, the frequency of cells resistant to apoptosis was highly variable. Moreover, resistant signaling states are not present before TRAIL treatment and are, therefore, not selected for. Instead, TRAIL induces different signaling states across cell lines and the overall change diversity upon treatment correlates with apoptotic resistance at the population level. By constricting the diversity of achievable TRAIL-induced signaling states with kinase inhibitors, we could also concomitantly decrease resistance and increase progression to apoptosis. This was directly associated with de novo protein synthesis in response to TRAIL that we demonstrated was required for apoptotic escape.

The mechanisms by which cancer cells successfully evade the action of cell death–inducing therapies ultimately lead to most cancer fatalities. The traditional understanding of resistance is where genetic aberrations confer a selective advantage on a subset of the population is insufficient to explain resistance (Cohen et al, 2008; Lee et al, 2012). Here, we add to the role of nongenetic mechanisms in driving resistance and the probability of multiple disparate mechanisms of resistance acting concurrently to confer resistance (Frank & Rosner, 2012). Integrating information on various sources of resistance will help us better model resistance to drugs such as TRAIL which is plagued by unexplainable clinical inefficacy. To date, resistance to TRAIL therapy is still debated in part because of the engagement of multiple canonical and noncanonical signaling pathways downstream of TRAIL. The complex interplay between these and the resulting heterogeneity in cellular and patient responses confounds this understanding further.

With this in mind, previous studies have taken clonal, single-cell approaches to establishing a more systemic resistance mechanism. For instance, preexisting variation in essential apoptotic proteins in individual cells due to stochasticity has been previously implicated in TRAIL resistance (Spencer et al, 2009). This supports the view that nongenetic resistance to TRAIL is driven by the Darwinian selection of preexisting cellular features such as protein levels. However, there is a growing body of literature that characterizes a Lamarckian induction of cellular features that allow a cell to become resistant upon exposure to the drug (Pisco et al, 2013; Fallahi-Sichani et al, 2017; Shaffer et al, 2017). Here, we were able to show for the first time that acquired, nongenetic mechanisms such as induction of survival signaling and variation in it upon TRAIL exposure allows different cell line models to variably persist and achieve a stable, TRAIL resistant state. This was uniquely possible through our utilization of highly multiplexed single-cell signaling assays to characterize the apoptotic response to TRAIL using mass cytometry (Bendall et al, 2011).

Variation in signaling is known to confer robustness at the population level and allow differential responses to the same stimuli (Schaefer et al, 2014). Therefore, it is logical to consider that the more diverse the signaling states present in a population, the more robust it is to environmental stimuli. We see this with increased induction of variation in signaling correlating with increased resistance to TRAIL at the population level. By changing the diversity of signaling states of response using kinase inhibitors, we showed that cells were concomitantly less robust and more sensitive toward TRAIL. We quantified high-dimensional heterogeneity in signaling states using Shannon diversity index, and we purport that we can use this index as a combination therapy design tool (Huang et al, 2017).

Given the crucial role of induced variation in TRAIL resistance, what might underlie and drive this diversity? We show that higher levels of de novo translation in TRAIL-resistant cells is directly associated with the variable TRAIL-induced signaling state. Thus, the requirement of translation for TRAIL resistance combined with the associated signaling variability implicates translation induced by TRAIL as a key contributor to the diversity in resistant states. A potential link between the population-level behavior in fractional TRAIL killing, and intracellular control of translation could be that viable population density might be influencing intracellular signaling pathways that promote translation. The relationship between cell density and drug resistance has been previously researched in other drugs, where the Hippo-yes-associated protein 1 pathway has been characterized as a molecular link (Pernicová et al, 2014; Gujral & Kirschner, 2017; von Manstein & Groner, 2017). Contact inhibition in TRAIL resistance and its link to translation is yet unexplored and might better inform models of TRAIL resistance.

Apart from translation, epigenetic heterogeneity is also likely driving differential signaling states. The epigenome controls chromatin accessibility to transcription machinery and subsequently influences protein synthesis and final cell state. The link between the chromatin accessibility profile and cell state is known, and therefore, preexisting variation in chromatin state could explain the variation in TRAIL-induced cell states (Lara-Astiaso et al, 2014). Poised chromatin states have been shown to lead to drug-tolerant reversible states with more prolonged exposure permanently altering the epigenome to allow for more stable resistance (Sharma et al, 2010; Brown et al, 2014). Single-cell technologies that capture chromatin accessibility profiles such as single-cell Assay for Transposase-Accessible Chromatin using sequencing could help explore the influence of epigenomic variability on the heterogeneity of resistance to cancer drug such as TRAIL (Buenrostro et al, 2015; Litzenburger et al, 2017). Still, our data indicate that this state cannot be selected for in the short term, suggesting that this epigenetically TRAIL-resistant state, if it exists, is part of a continuum of states whose abundance is cell type specific. Future work could investigate for common themes in regulatory epigenetic elements downstream of TRAIL-induced signaling.

Altogether, we applied a single cell, high-dimensional systems biology approach to study TRAIL resistance which led to us identifying diversity of signaling states as a new, conserved nongenetic mechanism of resistance to TRAIL. This nongenetic resistance study encourages further work to identify and understand its fundamental drivers and explore its role in other drugs and cell death inducing ligands beyond TRAIL.

# Materials and Methods

### Cell culture

HeLa C9, Ntera2, and MDAMB231 cells were cultured in DMEM containing 10% FBS and 1% penicillin/streptomycin (Gibco). NCIH460,

Jurkats, REH, U937, HEL, and HCC827 cells were cultured in RPMI containing 10% FBS, 5 mM L-glutamine and 1% penicillin/streptomycin (Gibco). HCT116 cells were cultured in McCoy's media containing 10% FBS and 1% penicillin/streptomycin (Gibco). The cells were treated with TRAIL when they were at 60–70% confluency to exclude density-dependent effects on resistance to TRAIL-induced apoptosis.

## Mass cytometry experiments

The cells were treated with SuperKiller TRAIL (Enzo Life Sciences; 50 ng/ml) for indicated times. In all experiments involving pertur-bagens, the cells were pretreated either with DMSO or with the following perturbagens 1 h before application of TRAIL: JNK Inhibitor I (EMD Millipore; 2 $\mu$M), Ku-0063794 (Selleck Chemicals; 1 $\mu$M), SB203580 (Cell Signaling Technology; 20 $\mu$M), and GDC-0941 (Selleck Chemicals; 2.5 $\mu$M).

The cells were treated with 10 $\mu$M IdU (Sigma-Aldrich) (Behbehani et al, 2012) and 10 $\mu$M puromycin (Kimmey et al, in press) for 30 min before harvesting. IdU is used to label dividing cells in the S phase as DNA replication occurs (Behbehani et al, 2012). Puromycin is used as label to tag nascent peptides in ribosomes for a single-cell translation rate tracking technique, which quantifies the level of puromycin per cell using a metal isotope-tagged monoclonal antibody to puromycin on mass cytometry (Kimmey et al, 2019).

To halt survival and apoptotic signaling, the cells were fixed with formaldehyde (PFA; Electron Microscopy Sciences) added directly to growth media at a final concentration of 1.6% for 10 min at room temperature and washed twice with staining media (PBS with 0.5% BSA, 0.02% sodium azide) to remove residual PFA. Cells were permeabilized with methanol for 10 min at 4°C, then optionally stored at –80°C for later use. The cells were then washed twice in cell staining media to remove remaining methanol and stained for intracellular proteins for 30 min at room temperature. Staining cocktails are listed in Table S1. The cells were washed with CSM and stained with 1 ml of 2000× Ir DNA intercalator (diluted 1:5,000 in PBS with 1.6% PFA; DVS Sciences) for 20 min at room temperature or overnight at 4°C. Before CyTOF analysis, the cells were washed once with CSM and then twice with ddH$_2$O.

## Clonogenic assays

Clonogenic assays were carried out in triplicate. Cells growing in log phase were plated in six-well plates at the following densities: HeLa cells—2,000 cells/well, HCT116 cells—10,000 cells/well, and NCIH460—1,000 cells/well under standard cell culture conditions. TRAIL or DMSO was added to the culture media 24 h after plating at the concentrations listed above, the plates were incubated in cell culture incubators for 1 h and TRAIL 50 ng/ml was added, and the plates were incubated in cell culture incubators for 24 h. The plates were then washed two times with 37° medium and the medium was refreshed. The cells were then cultured until each colony contained at least 50 cells. Colonies were stained with crystal violet and counted using a light box. For serial passage experiments, HeLa C9 cells were plated out in 25-cm$^2$ flasks at 4 × 10$^5$ cells and treated in parallel with clonogenic assays under identical conditions. Cells from flasks plates were then passaged two times over 10 d, plated

for clonogenic assays, re-treated with the same conditions, and analyzed.

## TRAIL receptor (DR4) flow cytometry experiments

Endogenous cell lines were grown to 60% confluency and harvested before fixing with 1.6% PFA for 10 min. Fixed cells were stained with 4 $\mu$g/ml of TRAIL receptor 1 primary antibody (Catalog number AF347 from R&D systems) and anti-goat secondary antibody conjugated to Alexa 488 (ab150129; Abcam) for 30 min each at room temperature. Titration data and unstained and secondary stained controls are shown in Fig S1.

## Mass cytometry analysis data preprocessing

To make all samples maximally comparable, data were acquired using internal metal isotope bead standards as previously described (Finck et al, 2013). Cell events were acquired at ~300 events per second on a CyTOF (DVS Sciences) using instrument settings previously described (Finck et al, 2013). Each sample was individually normalized to the internal bead standards before analysis. To remove post-apoptotic cells and debris, the cells were gated based on cell length and DNA content (Bendall et al, 2011). Mass cytometry dot plots, histograms, and heat maps were created either on www.cytobank.org or R with signal strength displayed on an arcsinh scale (the inverse hyperbolic sine) (Chen & Kotecha, 2014).

## viSNE analysis

All viSNE analyses were performed on Cytobank with equal sub-sampling of 20 K non-apoptotic cells per sample across the time course with standard settings of 1,000 iterations, perplexity of 30, and $\theta$ of 0.5 (Amir et al, 2013; Chen & Kotecha, 2014).

## Wanderlust analysis

Non-apoptotic HeLa C9 cells after 0–8 h of TRAIL treatment were equally subsampled to 5 K cells from each sample and concate-nated into one CSV file before building a linear trajectory based on the signaling markers using the Wanderlust implementation on MATLAB (Bendall et al, 2014a). The first derivative of output signaling changes was visualized along the calculated trajectory. The output CSV file had an extra column with a wanderlust score between 0 and 1 for every non-apoptotic cell from the pooled time course. The cells were binned according to their wanderlust score and visualized as a histogram on R.

## Signaling diversity calculation

The non-apoptotic cell populations of interest were gated on Cytobank and exported into R. 10 K cells from each file was sub-sampled and arcsinh transformed. 14 markers were selected (shown in Table S1) and the counts were scaled between 0 and 1. Shannon diversity index was then calculated over the data matrix using the Euclidean distance metric. The median diversity index per sample was then correlated to the calculated SQ of each cell line using Pearson correlation.

## Statistical work and visualizations in R

ggplot2 package was used for visualizing data with dot plots and boxplots. Correlation between markers on arcsinh-transformed data values was carried out using Spearman correlation with the DCGA package. Heat maps were made with heatmap.2 function from the gplots package. Mutual information (DREMI) scores and plots were created using the scprep stats toolkit (Krishnaswamy and Spitzer, 2014).

## Data availability

All live cell gated data in FCS files are publicly available on flowrepository.org with the following IDs: FR-FCM-Z276, FR-FCM-Z277, FR-FCM-Z278, FR-FCM-Z279, FR-FCM-Z27A, FR-FCM-Z27B, FR-FCM-Z27C, FR-FCM-Z27D and FR-FCM-Z27E.

## Supplementary Information

## Acknowledgements

R Baskar holds the A*STAR National Science Scholarship (PHD Doctor of Philosophyo) and is funded by A*STAR Singapore. S Kimmey is supported by the NIH/NIGMS (National Institute of Health/ National Institute of General Medical Sciences) Cell and Molecular Biology Training Grant (T32GM007276). SC Bendall is supported by the Damon Runyon Cancer Research Foundation Fellowship (DRG-2017-09), the NIH 1DP2OD022550-01, 1R01AG056287–01, 1R01AG057915-01, 1-R00-GM104148-01, 1U24CA224309-01, 5U19AI116484-02, U19 AI104209, The Bill & Melinda Gates Foundation, and a Translational Research Award from the Stanford Cancer Institute.

## Author Contributions

R Baskar: data curation, formal analysis, validation, investigation, visualization, methodology, and writing—original draft, review, and editing.
HG Fienberg: conceptualization and resources.
Z Khair: data curation, investigation, and methodology.
P Favaro: resources, data curation, and investigation.
S Kimmey: resources, investigation, and methodology.
DR Green: resources, formal analysis, and investigation.
GP Nolan: conceptualization, resources, supervision, funding acquisition, investigation, methodology, project administration, and writing—original draft, review, and editing.
S Plevritis: data curation, formal analysis, and investigation.
SC Bendall: conceptualization, data curation, supervision, funding acquisition, investigation, methodology, project administration, and writing—original draft, review, and editing.

## Conflict of Interest Statement

The authors declare that they have no conflict of interest.

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
