## [Reviewer comments · Life Science Alliance]

Life Science Alliance

TRAIL Induced Variation of Cell Signaling States Provides Non-Heritable Resistance to Apoptosis

Reema Baskar, Harris Fienberg, Zumana Khair, Patricia Favaro, Sam Kimmey, Douglas Green, Garry Nolan, Sylvia Plevritis, and Sean Bendall

DOI: <https://doi.org/10.26508/lsa.201900554>

Corresponding author(s): Sean Bendall, Stanford School of Medicine

Review Timeline:

Submission Date:	2019-09-16
Editorial Decision:	2019-09-16
Revision Received:	2019-09-18
Editorial Decision:	2019-10-10
Revision Received:	2019-10-24
Accepted:	2019-10-25

Scientific Editor: Andrea Leibfried

Transaction Report:

Please note that the manuscript was previously reviewed at another journal and the reports were taken into account in the decision-making process at Life Science Alliance.

Reviewer #1 Review

Report for Author:

The paper from the Bendall group presents an elegant mass cytometry study regarding non-genetic heterogeneity in response to TRAIL in 10 cell lines. By process of elimination, the authors concluded that genetic and cell cycle dependent mechanisms are not involved. The diversity of signaling states induced by TRAIL and protein synthesis were found to be correlated to the cell survival quotient. The cells would move on to a common signaling state prior to deciding whether to stay in that state or move on to a pro-apoptotic state. In a sense, this study lends support to the study by Spencer et al. back in 2009. While the study of TRAIL responses in cell lines is of moderate impact, the data presented are of high quality.

Major concerns

1. It is unclear how many independent replicates are performed for each mass cytometry experiment and subsequent survival quotient (SQ) and entropy calculations. I was expecting that there was barcoding and pooling of multiple replicates, which can later be deconvoluted. However, none of the figure legends or methods mentioned n . We assume n is greater than 1, (i.e., n cannot be the number of cells in a single run). Increasing n would speak to the relative robustness of signaling mechanisms.
2. Diversity of signaling correlating to resistance phenotypes. The authors concluded that the entropy measure (the diversity or variance of signaling states) is correlated to a survival quotient. However, variance is usually correlated to the amplitude of a signal even if the data are variance-corrected. Thus, the mechanism can be: that increased amplitude of signaling in multiple pathways, where many of the markers measured are in survival pathways, is correlated to survival. This mechanism will still be consistent with the inhibitor results (which the authors claimed to decrease heterogeneity and signaling magnitude at the same time). Please discern these hypotheses.
3. Mechanisms of resistance. The mechanism of adaptive resistance is still not conclusively discerned, and could still be explained by some degree of heterogeneity before TRAIL treatment. For instance, it is unclear whether resistant cell states exist prior to treatment or is it a truly Lamarckian process by which a group of cells adapt dynamically. Some of these mechanisms can only be functionally tested. For instance, the authors excluded the possibility of cell cycle effects. Can there be an experiment done where cell cycle of cells is synchronized prior to treatment?
4. With regards to the low entropy of untreated cells, this again can be due to the amplitude argument mentioned in point 2. I would again not discount the existence of heterogeneity prior to treatment, just because it is not detectable or measurable (differences below the sensitivity of detection). These cells may be functionally heterogeneous and respond differently).
5. Why are cells able to exhibit diversity in protein translation? If it is a contact dependent mechanism, a marker can certainly be identified in densely populated versus sparsely populated area in a cell line. Furthermore, if you hit the cells in a cell cycle phase where translation is active, are cells then more likely to resist treatment? The heterogeneity in protein translation should be further explored
6. .
7. The concept of entropy probably needs to be moderated and be made specific to the markers used in the study since it is only calculated from selected markers measured in the panel. It cannot be excluded that there may be the same amount or even less entropy if the entire signaling space of the cell is considered.

Minor concerns

1. Figure S2 legend is incorrect
2. Kimmey et al. paper in press is not in citations
3. The methods mentioned patient specimens. There are no patient specimens in the paper.

Reviewer #2 Review

Report for Author:

In their Manuscript "TRAIL induced variation of cell signalling states provides non-heritable resistance to apoptosis", the authors use mass cytometry to investigate how signalling states of

cells relate to TRAIL resistance. They find that heterogeneity in signalling states induced by TRAIL correlates with TRAIL resistance, and increased translation, presumably induced by mTOR/S6 is a mechanism of TRAIL resistance.

The authors use 10 different cell lines and show that response to TRAIL is variable, with also variability on signalling response (exemplified by S6). Using restimulation experiments, the authors show that survival/resistance is not maintained and rather a stochastic event, and CyTOF analysis shows that the signalling response of pre-exposed and control are similar. The authors then show that signalling heterogeneity (entropy) after TRAIL treatment correlates with resistance and changes in heterogeneity by inhibitors also show correlation with survival. Finally, the authors show that translation induction is one of the key features that leads to survival.

Overall I like the approach and also think that the results are important and well justified. However, I have a number of concerns as follows.

Major:

- As far as I understand the entire story builds on no replicates (n=1). At least the key experiments should be repeated.
- The authors attribute the increase in S6 to AKT/mTOR signalling. While this is likely, why not show it by using inhibitors. In the moment this statement is contradicted by experiments shown in Fig. 5, where S6 seems to be unaffected by PI3K/mTOR inhibitors (I find it also puzzling that pAKT increases when PI3K signalling is blocked, see Fig. 5D).
- The authors show that the TNF receptor mRNA levels don't correlate with the response. This should be shown on the surface protein level, as it is unclear how the mRNA levels of these receptor correlate with functional protein on the surface.

Minor:

- Fig 1B: I have a problem with the term emergence of "late stage cCasp_low cPARP_high apoptotic population" as the same population also exist at an earlier timepoint (2hrs). Maybe there is just some high variability?
- While I like the idea about cellular entropy, I think it is rather poorly defined and difficult to interpret. Why not use (the sum of) coefficient of variation or something similar, which are related to entropy but much better defined and easier to digest? Would also be important to understand which combination of marker variability is really decisive.

Reviewer #3 Review

Summary

Baskar et. al. use mass cytometry to characterize heterogeneous responses of different cancer cell lines to the apoptosis inducing ligand, TRAIL. This work builds upon previous reports of TRAIL-induced fractional killing in cancer cell lines that show non-genetic cell variability in the apoptotic response. Using a panel of antibodies for various proteins involved in apoptosis, cell cycle, and

survival signaling, the authors hoped to identify a TRAIL-induced transition state that all cells occupy before selectively progressing to apoptosis. As a measure of such a transition state, the authors compute an entropy-based metric that correlates heterogeneity changes with cell survival rates. Further, they find that several different survival signaling-related kinases and targets in the CyTOF panel are induced in cells that are resistant to TRAIL. Adding puromycin to the CyTOF panel, the authors claim that cell-to-cell variability depends on de novo TRAIL-induced translation in treated cells.

General remarks

Overall, the manuscript overstates the findings and the data do not support claims made by the authors regarding specific TRAIL-induced changes in cells. There are previous reports to justify the authors' interests in fractional killing of cancer cells by TRAIL (PMID: 19363473, 22108795, 22570596, 23699397, etc.), and this work does little that goes beyond the earlier studies. The authors generated a high-dimensionality dataset by measuring 30 proteins in single cells for 10 cancer cell lines, both pre- and post-treatment with TRAIL. While these data provide a nice resource for future systems models of heterogeneity in TRAIL response, the manuscript here is largely phenomenological. Follow-on experiments and analyses lack depth and do not add to the authors' claims. While initial measurements are made for all 10 cell lines, findings seem cell-type specific and most of the findings are only followed up in one cell line. These general critiques are expanded upon later in the review.

This manuscript seems more appropriate for a proteomics-focused journal that allows a resource type contribution. It does not meet the criteria or scientific scope of Molecular Systems Biology. I suggest the authors focus their writing on motivating and describing the experimental design for this dataset to make it accessible for those interested in developing models that are dependent on cell-type context and cell-to-cell variability, such as models coupling cell-cycle progression and apoptosis.

Major points

Critique of this manuscript falls under two major categories that are elaborated upon below. For each, a non-exhaustive list of representative examples is provided.

There is a repeated lack of transparency in the data presented in several figures and accompanying text:

- The actual numerical data generated by mass cytometry not provided anywhere in the manuscript. Every heatmap is accompanied by a scale bar ranging from "minimum" to "maximum", lacking actual numerical ranges and thereby obscuring the data. While there is a table in the supplement listing the antibodies used, there is no cataloging of experimental conditions and data generated.
- The authors use different gating parameters for "viable" and "non-apoptotic" cells that are never described and vary between figures. How the two different gates are deployed is also unclear.
- The authors use 10 cell lines but do not describe the differences between them. For example, both "Hela" and "Hela C9" cells are used with the latter not having any literature references nor a description. As most of their follow-on analyses are in Hela C9 cells, the lack of description of this Hela derivative cell line is notable.
- There is discrepancy in data between different figures for the same cell line and experimental measurements that are not addressed by the authors. For example, the "survival quotient"-a critical readout used by the authors for estimating TRAIL resistance-varies by 1.5-2x for the same cell line (Hela C9) in the same treatment condition between Fig. 1D and Fig. 5B.

There are several instances where the data presented in the figures are misinterpreted in the text,

which confounds the conclusions made by the authors:

- In Fig. 5, the authors attempt to perturb resistance to TRAIL treatment by inhibiting a variety of kinases. However, nowhere in the main figure or the supplement is the inhibitory action of the molecules verified. In fact, the heatmap in Fig. 5D suggests that these inhibitors do not work as described by the authors. For example, pAKT increases in response to inhibition of its activator PI3K and pS6 increases in response to inhibition of its activator mTOR. Both of these findings are opposite of what is expected but described by the authors as "compensatory signaling changes" without proper justification (Page 10). It seems that either the heatmap scale bar is reversed or misinterpreted or these inhibitors are not effective in this experimental design.
- In Fig. 6, the authors claim that resistance to TRAIL is dependent on treatment-induced increased translation, and inhibiting translation with cycloheximide abrogates resistance. However, in Fig. 6B we see that the population of cells with high translation rate is unaffected by cycloheximide alone. It is only when combined with TRAIL that cycloheximide treatment results in decrease in translation rate. The authors conclude that this implies a causative relationship between overall translation and TRAIL resistance. However, the data suggest that TRAIL treatment sensitizes cells to translation inhibition by cycloheximide. The manuscript does not even consider the well known explanation that translation of NF-kappaB-induced transcripts will promote survival in TRAIL-treated cells.
- While the major technical advance of this manuscript is in its high dimensional single-cell measurements, the authors do not adequately explore single-cell correlations of different protein markers. While overall heterogeneity is characterized in Fig. 3D and Fig. 4C, actual molecular variations are not detailed. Within a single cell-line, a 20x20 correlation plot of changes in different proteins in response to TRAIL would offer a view of the entire dataset, which would allow the reader to evaluate whether cell lines have coordinated or divergent responses to TRAIL treatment.

Minor points

- Consistent numerical ranges on heatmaps are necessary.
- The re-treatment experiment conditions in Fig. 2 would be better renamed with + and - signs. For example, N1T2 is better shown as -/+.
- There are numerable typographical errors including allusions to "patient samples" in the methods section, as well as incorrect figure captions in the supplement.

September 16, 2019

Re: Life Science Alliance manuscript #LSA-2019-00554-T

Dr Sean C Bendall
Stanford School of Medicine
Pathology
3373 Hillview Ave
Room 230a
Palo Alto, CA 94304

Dear Dr. Bendall,

Thank you for transferring your manuscript entitled "TRAIL Induced Variation of Cell Signaling States Provides Non-Heritable Resistance to Apoptosis" to Life Science Alliance. The manuscript was assessed by expert reviewers at another journal before, and the editors transferred those reports to us with your permission.

The reviewers appreciated the quality of your data but would have expected a further reaching advance. This is not precluding publication in Life Science Alliance, and we would thus like to invite you to submit a revised version of your manuscript to us.

As already outlined to you in our discussion prior to submission, we would expect a full point-by-point response and accordingly changes to the manuscript text to discuss alternative hypothesis (rev#1), addition of information, clarifications and toning-down of the conclusions (all reviewers), making sure of having several replicates (rev#1 and 2), fixing or removing the inhibitor data that gave perplexing outcomes (rev#2 and 3), analysis of TNFR surface abundance (rev#2), adding more insight into the translation part (rev#3).

Thank you for this interesting contribution to Life Science Alliance. We are looking forward to receiving your revised manuscript.

Sincerely,

Andrea Leibfried, PhD
Executive Editor
Life Science Alliance

Meyerhofstr. 1
69117 Heidelberg, Germany
t +49 6221 8891 502
e a.leibfried@life-science-alliance.org
www.life-science-alliance.org

B. MANUSCRIPT ORGANIZATION AND FORMATTING:

Reviewers' comments

Author responses

Reviewer #1:

The paper from the Bendall group presents an elegant mass cytometry study regarding non-genetic heterogeneity in response to TRAIL in 10 cell lines. By process of elimination, the authors concluded that genetic and cell cycle dependent mechanisms are not involved. The diversity of signaling states induced by TRAIL and protein synthesis were found to be correlated to the cell survival quotient. The cells would move on to a common signaling state prior to deciding whether to stay in that state or move on to a pro-apoptotic state. In a sense, this study lends support to the study by Spencer et al. back in 2009. While the study of TRAIL responses in cell lines is of moderate impact, the data presented are of high quality.

We thank the reviewer for the feedback on data quality and agree with them that these data presented here both extend and reconcile the observations around TRAIL induced apoptosis heterogeneity by Spencer et al. in Nature.

Major concerns

1. It is unclear how many independent replicates are performed for each mass cytometry experiment and subsequent survival quotient (SQ) and entropy calculations. I was expecting that there was barcoding and pooling of multiple replicates, which can later be deconvoluted. However, none of the figure legends or methods mentioned n. We assume n is greater than 1, (i.e., n cannot be the number of cells in a single run). Increasing n would speak to the relative robustness of signaling mechanisms.

The signaling diversity was calculated for each cell line with 1-4 independent biological replicates and this information is better reflected in the figures (example, fig 4E-F below) and figure legends of the revised manuscript.

2. Diversity of signaling correlating to resistance phenotypes. The authors concluded that the entropy measure (the diversity or variance of signaling states) is correlated to a survival quotient. However, variance is usually correlated to the amplitude of a signal even if the data are variance-corrected. Thus, the mechanism can be: that increased amplitude of signaling in multiple pathways, where many of the markers measured are in survival pathways, is correlated to survival. This mechanism will still be consistent with the inhibitor results (which the authors claimed to decrease heterogeneity and signaling magnitude at the same time). Please discern these hypotheses.

The reviewer raises an interesting point about our original analysis, to account for this we have included a simplified analysis that accounts for signaling amplitude in the revised manuscript. The entropy metric was simplified to Shannon's diversity index calculated directly on arsinh -transformed signaling marker values re-scaled to 0-1 (the revised methods on signaling diversity calculations contains detailed information). This diversity calculation is not impacted by the absolute abundance of the markers and only assesses the relative differences in the scaled signaling marker values in comparison to each other in each cell of the equally subsampled non-apoptotic population. Shannon's diversity index is a well-known metric that captures the heterogeneity present in single cell biology and by normalizing the data beforehand, we ensured that any differences in the absolute abundance of the markers did not confound signaling diversity of the cells (Chung et al., 2017; Park et al., 2010). With this metric we reinforce our original conclusion and show that the signaling diversity of the resistant cells across cell lines and inhibitor treatments significantly correlate with resistance (survivor quotient) (Figure 4F, 5F, revised manuscript).

3. Mechanisms of resistance. The mechanism of adaptive resistance is still not conclusively discerned, and could still be explained by some degree of heterogeneity before TRAIL treatment. For instance, it is unclear whether resistant cell states exist prior to treatment or is it a truly Lamarckian process by which a group of cells adapt dynamically. Some of these mechanisms can only be functionally tested. For instance, the authors excluded the possibility of cell cycle

effects. Can there be an experiment done where cell cycle of cells is synchronized prior to treatment?

This is an interesting question about resistance in general raised by the reviewer. Due to the nature of our multiplexed assessment of apoptosis, the same cells are not tracked from before TRAIL treatment and throughout the time-course. Therefore, we cannot truly ascertain whether the same resistant cells existed prior to treatment.

However, we show that: 1) virtually all cells signal in response to TRAIL, 2) there is a wide dichotomy of signaling states cells achieve are very different in their signaling states which are likely TRAIL-induced. We are also able to capture the cell cycle states of cells using 4 key readouts; IdU, cyclinB1 and phosphorylated H3 and Rb (Behbehani et al., 2012). These markers delineate S, G1/G2, M and G0 phase respectively. We also further have a pan proliferation marker Ki67 to marker cells actively dividing. As we have the cell cycle status of all cells across cell lines, we were able to quantitatively ascertain the cell cycle differences after TRAIL treatment and felt this was sufficient to show that TRAIL-induced cell cycle changes were cell type specific – effectively we were able to synchronize the cell cycle to each stage in silico (Figure 3B, revised manuscript). Due to the lack of conserved cell cycle differences with TRAIL treatment across cell lines observed here, we believed there was little additional value in physically/chemically synchronizing cell cycle in additional experiments – which can also further confound apoptotic readouts.

4. With regards to the low entropy of untreated cells, this again can be due to the amplitude argument mentioned in point 2. I would again not discount the existence of heterogeneity prior to treatment, just because it is not detectable or measurable (differences below the sensitivity of detection). These cells may be functionally heterogeneous and respond differently).

Again, we simplified the high dimensional signaling diversity calculation to Shannon's diversity index calculated on unit scaled data for each sample individually, removing the effect of signaling amplitude differences on our entropy calculations in the revised manuscript.

We agree with the reviewer that it is likely that heterogeneity exists within the starting population of cells. We believe this functional heterogeneity is revealed here by TRAIL treatment and the resulting differential signaling responses and apoptotic resistance. However, we do not see a conserved trend with resistance (survivor quotient) across our experiments in untreated non-apoptotic cells (Figure 4F, revised manuscript). We observe significant positive correlation only when we consider treated non-apoptotic cells, across cell lines and combination therapies. So, for the purpose of this study, we put forward signaling diversity in response to TRAIL, likely an induced phenomenon, as a mechanism of non-genetic resistance to TRAIL.

5. Why are cells able to exhibit diversity in protein translation? If it is a contact dependent mechanism, a marker can certainly be identified in densely populated versus sparsely populated area in a cell line. Furthermore, if you hit the cells in a cell cycle phase where translation is active, are cells then more likely to resist treatment? The heterogeneity in protein translation should be further explored

In this case the diversity in translation was linked directly to the diversity in TRAIL induced signaling. If we modulated signaling with chemical inhibitors, we also change the de novo protein translation response and if we reduced (via sublethal cycloheximide) de novo protein translation we reduced resistance to TRAIL induced apoptosis. We have added figures and additional explanation in the results section of the revised manuscript to better describe how translation and the non-canonical signaling response to TRAIL are linked together in our data (Figure 6 & S6E). Given the strong correlation between the two it is likely these two events are linked. For instance, these processes could be working together through a positive feedback loop and additionally explain the heterogeneity in protein translation.

6. The concept of entropy probably needs to be moderated and be made specific to the markers used in the study since it is only calculated from selected markers measured in the panel. It cannot be excluded that there may be the same amount or even less entropy if the entire signaling space of the cell is considered.

We agree with the reviewer and moderated our concept of entropy by changing it to a simple Shannon's diversity index calculation over 14 key markers in our panel. This certainly does not capture the entire signaling space of the cell, but it tries to capture the diversity in the key signaling pathways known to be downstream of TRAIL.

Minor concerns

1. Figure S2 legend is incorrect

Corrected

2. Kimmey et al. paper in press is not in citations

Corrected

3. The methods mentioned patient specimens. There are no patient specimens in the paper.

Removed from the revised manuscript

Reviewer #2:

In their Manuscript "TRAIL induced variation of cell signalling states provides non-heritable resistance to apoptosis", the authors use mass cytometry to investigate how signalling states of cells relate to TRAIL resistance. They find that heterogeneity in signalling states induced by TRAIL correlates with TRAIL resistance, and increased translation, presumably induced by mTOR/S6 is a mechanism of TRAIL resistance.

The authors use 10 different cell lines and show that response to trail is variable, with also variability on signalling response (exemplified by S6). Using restimulation experiments, the authors show that survival/resistant is not maintained and rather a stochastic event, and CyTOF analysis shows that the signalling response of pre-exposed and control are similar. The authors then show that signalling heterogeneity (entropy) after TRAIL treatment correlates with

resistance and changes in heterogeneity by inhibitors also show correlation with survival. Finally, the authors show that translation induction is one of the key features that leads to survival.

Overall I like the approach and also think that the results are important and well justified. However, I have a number of concerns as follows.

We thank the reviewer for the succinct summary of the approach and key findings and for the overall positive feedback on the methods and interpretation.

Major:

- As far as I understand the entire story builds on no replicates (n=1). At least the key experiments should be repeated.

While some of the qualitatively described time course experiments were n=1 as presented here, they were repeated across numerous cell line as well as the data presented here is based on dozens of optimization experiments yielding similar trends. The key signaling diversity experiments here for instance had data calculated for each cell line based on 1-4 independent biological replicates and this information is better reflected in the relevant figures (Figure 4E-F). Overall, the replicates have been better described in the figures of the corresponding data in the revised manuscript.

- The authors attribute the increase in S6 to AKT/mTOR signalling. While this is likely, why not show it by using inhibitors. In the moment this statement is contradicted by experiments shown in Fig. 5, where S6 seems to be unaffected by PI3K/mTOR inhibitors (I find it also puzzling that pAKT increases when PI3K signalling is blocked, see Fig. 5D).

This was not actually the case in the original manuscript. However, the issue was that untreated non-apoptotic cells were used as the baseline which hid the inhibitor-driven signaling changes – accounting for the reviewer's interpretation here. We rectified this by setting cells treated with 6h of TRAIL only as the baseline and improved the visualization of the data. We now show that phosphorylation of S6 is reduced by mTOR/PI3K inhibitors and pAKT decreases with PI3K inhibition when combination therapy samples are compared to 6h TRAIL only treated samples (Figure 5D, revised manuscript).

- The authors show that the TNF receptor mRNA levels don't correlate with the response. This should be shown on the surface protein level, as it is unclear how the mRNA levels of these receptor correlate with functional protein on the surface.

We have added new experimental data to Figure 1 where we captured the levels of TRAIL receptor 1 (Death receptor 4) protein on endogenous cell lines using flow cytometry (Figure 1E). We show that interestingly with higher DR4, there is greater resistance to TRAIL-induced apoptosis across cell lines (Figure 1F). Furthermore, we also show that mRNA levels of TNFRSF10A gene is very strongly correlated to DR4 protein levels across cell lines (Figure 1D). These results reinforce our original conclusion that TRAIL receptor availability seems to

be unrelated apoptotic resistance. Again, this is also in the background of all cells tested initially signaling in response to TRAIL.

Minor:

- Fig 1B: I have a problem with the term emergence of "late stage cCasp_low cPARP_high apoptotic population" as the same population also exist at an earlier timepoint (2hrs). Maybe there is just some high variability?

We agree with the reviewer and removed this from the figure and text.

- While I like the idea about cellular entropy, I think it is rather poorly defined and difficult to interpret. Why not use (the sum of) coefficient of variation or something similar, which are related to entropy but much better defined and easier to digest? Would also be important to understand which combination of marker variability is really decisive.

We agree with the reviewer and simplified our entropy measurement to Shannon's diversity index calculated on unit scaled signaling data from non-apoptotic cells. This provides a robust measure of signaling heterogeneity that can be comparable across cell lines, treatments and experiments.

Reviewer #3:

Summary

Baskar et. al. use mass cytometry to characterize heterogeneous responses of different cancer cell lines to the apoptosis inducing ligand, TRAIL. This work builds upon previous reports of TRAIL-induced fractional killing in cancer cell lines that show non-genetic cell variability in the apoptotic response. Using a panel of antibodies for various proteins involved in apoptosis, cell cycle, and survival signaling, the authors hoped to identify a TRAIL-induced transition state that all cells occupy before selectively progressing to apoptosis. As a measure of such a transition state, the authors compute an entropy-based metric that correlates heterogeneity changes with cell survival rates. Further, they find that several different survival signaling-related kinases and targets in the CyTOF panel are induced in cells that are resistant to TRAIL. Adding puromycin to the CyTOF panel, the authors claim that cell-to-cell variability depends on de novo TRAIL-induced translation in treated cells.

General remarks

Overall, the manuscript overstates the findings and the data do not support claims made by the authors regarding specific TRAIL-induced changes in cells. There are previous reports to justify the authors interests in fractional killing of cancer cells by TRAIL (PMID: 19363473, 22108795, 22570596, 23699397, etc.), and this work does little that goes beyond the earlier studies. The authors generated a high-dimensionality dataset by measuring 30 proteins in single cells for 10 cancer cell lines, both pre- and post-treatment with TRAIL. While these data provide a nice resource for future systems models of heterogeneity in TRAIL response, the manuscript here is largely phenomenological. Follow on experiments and analyses lack depth and do not add to the authors' claims. While initial measurements are made for all 10 cell lines, findings seem cell-type specific and most of the findings are only followed up in one cell line. These general critiques are expanded upon later in the review.

This manuscript seems more appropriate for a proteomics-focused journal that allows a resource type contribution. It does not meet the criteria or scientific scope of Molecular Systems Biology. I suggest the authors focus their writing on motivating and describing the experimental design for this dataset to make it accessible for those interested in developing models that are dependent on cell-type context and cell-to-cell variability, such as models coupling cell-cycle progression and apoptosis.

We thank the reviewer for an excellent summary of our study design. While we agree that the data and conclusions presented here reconcile well with published literature – we disagree that there is no additional contribution. All of the previous studies cited utilized far less diversity in biological systems interrogated. Furthermore, when a diverse number of systems is queried – shown here for the first time – we highlight that despite technical consistency there is much biological diversity in TRAIL signaling responses when focusing on apoptotic resistance specifically. This highlights the intractable problem of TRAIL resistance and provides the most comprehensive map of these resistance signaling events with single cell resolution to date.

Still, with that said, we have taken into consideration the reviewers' comments and refocused the revised manuscript on better describing the experimental construction and will be providing the signaling data as a resource with publication.

Major points

Critique of this manuscript falls under two major categories that are elaborated upon below. For each, a non-exhaustive list of representative examples is provided.

There is a repeated lack of transparency in the data presented in several figures and accompanying text:

- The actual numerical data generated by mass cytometry not provided anywhere in the manuscript. Every heatmap is accompanied by a scale bar ranging from "minimum" to "maximum", lacking actual numerical ranges and thereby obscuring the data. While there is a table in the supplement listing the antibodies used, there is no cataloging of experimental conditions and data generated.

We added the numerical ranges to all scales in the figures for greater clarity and we uploaded all the data generated to flowrepository.org. The data is available at repository IDs: FR-FCM-Z276, FR-FCM-Z277, FR-FCM-Z278, FR-FCM-Z279, FR-FCM-Z27A, FR-FCM-Z27B, FR-FCM-Z27C, FR-FCM-Z27D and FR-FCM-Z27E.

- The authors use different gating parameters for "viable" and "non-apoptotic" cells that are never described and vary between figures. How the two different gates are deployed is also unclear.

We and made the gates and their names consistent, with cells gated on live/dead viability stain to be called "live" and then live cells gated on cleaved caspase 3 and cleaved PARP to be called "non-apoptotic". We show the gating strategy in Figure 1B of the revised manuscript.

- The authors use 10 cell lines but do not describe the differences between them. For example,

both "Hela" and "Hela C9" cells are used with the latter not having any literature references nor a description. As most of their follow-on analyses are in Hela C9 cells, the lack of description of this Hela derivative cell line is notable.

This clone of Hela cells was obtained from the Sorger lab where single cell clones were generated by serial dilution followed by expansion and testing for differential TRAIL sensitivity (Flusberg & Sorger, 2013). This information as well as reference has been added to the methods section of the revised manuscript.

- There is discrepancy in data between different figures for the same cell line and experimental measurements that are not addressed by the authors. For example, the "survival quotient"-a critical readout used by the authors for estimating TRAIL resistance-varies by 1.5-2x for the same cell line (Hela C9) in the same treatment condition between Fig. 1D and Fig. 5B.

We thank the reviewer for picking up a mistake on our part in using a wrong gate to determine the non-apoptotic cell population. We have since used a more appropriate gate that captures the non-apoptotic population as shown below (left: Non-apoptotic gate on untreated Hela c9 cells in the combination inhibitor treatment experiments, right: Non-apoptotic gate on 6h TRAIL treated Hela c9 live cells). While certainly more accurate this adjustment did not materially change the quantified data or conclusion.

There are several instances where the data presented in the figures are misinterpreted in the text, which confounds the conclusions made by the authors:

- In Fig. 5, the authors attempt to perturb resistance to TRAIL treatment by inhibiting a variety of kinases. However, nowhere in the main figure or the supplement is the inhibitory action of the molecules verified. In fact, the heatmap in Fig. 5D suggests that these inhibitors do not work as described by the authors. For example, pAKT increases in response to inhibition of its activator PI3K and pS6 increases in response to inhibition of its activator mTOR. Both of these findings are opposite of what is expected but described by the authors as "compensatory signaling changes" without proper justification (Page 10). It seems that either the heatmap scale bar is reversed or misinterpreted or these inhibitors are not effective in this experimental design.

This was not actually the case in the original study, but we realize how the data presentation made this unclear. The heatmap in the original manuscript was set to untreated cells as the baseline which hid the signaling changes due to combination treatment with TRAIL. We fixed

this by setting 6h TRAIL treated cells as the baseline and this shows that the inhibitors change the levels of pS6 and pAKT as expected (Figure 5D, revised manuscript).

• In Fig. 6, the authors claim that resistance to TRAIL is dependent on treatment-induced increased translation, and inhibiting translation with cycloheximide abrogates resistance. However, in Fig. 6B we see that the population of cells with high translation rate is unaffected by cycloheximide alone. It is only when combined with TRAIL that cycloheximide treatment results in decrease in translation rate. The authors conclude that this implies a causative relationship between overall translation and TRAIL resistance. However, the data suggest that TRAIL treatment sensitizes cells to translation inhibition by cycloheximide. The manuscript does not even consider the well-known explanation that translation of NF-kappaB-induced transcripts will promote survival in TRAIL-treated cells.

We agree that the scenario described by the review is a definite possibility that is consistent with our observations and conclusions here. We don't see a conserved signaling response to TRAIL across cell lines and additionally NFkB is not activated/phosphorylated in response to TRAIL in all resistant cells, however, we do see that regardless of cell line and combination treatments, resistant cells show higher rates of translation (as indicated by puromycin levels)(Arcsinh ratio of indicated markers in 6h TRAIL treated cells when compared to untreated cells, data shown below). Therefore, while the described scenario could certainly be happening in some cases, it is not conserved and the more general take away and what our data highlights for the first time is a more complex relationship between cellular response to TRAIL and translation. Please note, at the chosen cycloheximide dose, there is no direct significant effect on translation (Figure 6A, revised manuscript), however with TRAIL treatment the signaling response is shown to co-occur with high levels of translation (as shown by higher levels of our single cell translation reporter, puromycin) (Figure 6C & S6E, revised manuscript). We have tempered our conclusions in the main text and provided a more thorough discussion of the data and its interpretations in the revised manuscript.

• While the major technical advance of this manuscript is in its high dimensional single-cell measurements, the authors do not adequately explore single-cell correlations of different protein markers. While overall heterogeneity is characterized in Fig. 3D and Fig. 4C, actual molecular variations are not detailed. Within a single cell-line, a 20x20 correlation plot of changes in different proteins in response to TRAIL would offer a view of the entire dataset, which would

allow the reader to evaluate whether cell lines have coordinated or divergent responses to TRAIL treatment.

To the reviewers comments, and in an effort to increase the accessibility of the cell signaling as a resource here we have added substantial new analysis methods to leverage on our high dimensional single cell data, including differential correlation plots of signaling markers between 2 experimental conditions and mutual information DREMI plots to examine in more detail the signaling protein relationship changes with treatment and combination therapy (Krishnaswamy et al., 2014)(Figure 5G & 5H, revised manuscript).

Minor points

- Consistent numerical ranges on heatmaps are necessary.
- The re-treatment experiment conditions in Fig. 2 would be better renamed with + and - signs. For example, N1T2 is better shown as -/+.
- There are numerable typographical errors including allusions to "patient samples" in the methods section, as well as incorrect figure captions in the supplement.

All minor concerns have been addressed (corrected or removed as per reviewer comments) in the main text and figures.

References:

- Behbehani, G. K., Bendall, S. C., Clutter, M. R., Fantl, W. J., & Nolan, G. P. (2012). Single-cell mass cytometry adapted to measurements of the cell cycle. *Cytometry. Part A : The Journal of the International Society for Analytical Cytology*, 81(7), 552–566. <https://doi.org/10.1002/cyto.a.22075>
- Chung, Y. R., Kim, H. J., Kim, Y. A., Chang, M. S., Hwang, K.-T., & Park, S. Y. (2017). Diversity index as a novel prognostic factor in breast cancer. *Oncotarget*, 8(57), 97114–97126. <https://doi.org/10.18632/oncotarget.21371>
- Flusberg, D. A., & Sorger, P. K. (2013). Modulating cell-to-cell variability and sensitivity to death ligands by co-drugging. *Physical Biology*, 10(3), 035002. <https://doi.org/10.1088/1478-3975/10/3/035002>
- Krishnaswamy, S., Spitzer, M. H., Mingueneau, M., Bendall, S. C., Litvin, O., Stone, E., ... Nolan, G. P. (2014). Systems biology. Conditional density-based analysis of T cell signaling in single-cell data. *Science (New York, N.Y.)*, 346(6213), 1250689. <https://doi.org/10.1126/science.1250689>
- Park, S. Y., Gönen, M., Kim, H. J., Michor, F., & Polyak, K. (2010). Cellular and genetic diversity in the progression of in situ human breast carcinomas to an invasive phenotype. *The Journal of Clinical Investigation*, 120(2), 636–644. <https://doi.org/10.1172/JCI40724>

October 10, 2019

RE: Life Science Alliance Manuscript #LSA-2019-00554-TR

Prof. Sean C Bendall
Stanford School of Medicine
Pathology
3373 Hillview Ave
Room 230a
Palo Alto, CA 94304

Dear Dr. Bendall,

Thank you for submitting your revised manuscript entitled "TRAIL Induced Variation of Cell Signaling States Provides Non-Heritable Resistance to Apoptosis". As you will see, original reviewer #2 appreciates the introduced changes and we would thus be happy to publish your paper in Life Science Alliance pending final revisions necessary to meet our formatting guidelines:

- Please link your profile in our submission system to your ORCID iD, you should have received an email with instructions on how to do so
- Please provide the manuscript text in docx format
- Please list 10 authors et al in the reference list
- Please upload all figures (also suppl. Figures) as individual files; the figure legends should remain in the main ms file and the supplementary figure legends and table S1 can get moved into the main ms file as well, please
- Please note that we display the suppl figures in-line in the HTML version of the paper; please split Figure S4 into two figures
- Please add callouts in the ms text to Fig S1E and S5B

A. FINAL FILES:

B. MANUSCRIPT ORGANIZATION AND FORMATTING:

Sincerely,

Andrea Leibfried, PhD
Executive Editor
Life Science Alliance
Meyerhofstr. 1
69117 Heidelberg, Germany

t +49 6221 8891 502
e a.leibfried@life-science-alliance.org
www.life-science-alliance.org

Reviewer #2 (Comments to the Authors (Required)):

Very sorry for the late response, I was overbusy during the last weeks. I think the authors adressed all of my concerns, and I recommend publication.

October 25, 2019

RE: Life Science Alliance Manuscript #LSA-2019-00554-TRR

Prof. Sean C Bendall
Stanford School of Medicine
Pathology
3373 Hillview Ave
Room 230a
Palo Alto, CA 94304

Dear Dr. Bendall,

Thank you for submitting your Research Article entitled "TRAIL Induced Variation of Cell Signaling States Provides Non-Heritable Resistance to Apoptosis". It is a pleasure to let you know that your manuscript is now accepted for publication in Life Science Alliance. Congratulations on this interesting work.

DISTRIBUTION OF MATERIALS:

Again, congratulations on a very nice paper. I hope you found the review process to be constructive and are pleased with how the manuscript was handled editorially. We look forward to future exciting submissions from your lab.

Sincerely,

Andrea Leibfried, PhD
Executive Editor
Life Science Alliance
Meyerohofstr. 1
69117 Heidelberg, Germany
t +49 6221 8891 502
e a.leibfried@life-science-alliance.org
www.life-science-alliance.org